# Deep learning enables satellite-based monitoring of large populations of terrestrial mammals across heterogeneous landscape

Zijing Wu [1,14], Ce Zhang [2,3,14], Xiaowei Gu [4,14], Isla Duporge [5,6,7,14], Lacey F. Hughey [8], Jared A. Stabach [8], Andrew K. Skidmore [1,9], J. Grant C. Hopcraft [10], Stephen J. Lee[6], Peter M. Atkinson[2,11], Douglas J. McCauley[12], Richard Lamprey [1], Shadrack Ngene[13] & Tiejun Wang [1] ✉

New satellite remote sensing and machine learning techniques offer untapped possibilities to monitor global biodiversity with unprecedented speed and precision. These efficiencies promise to reveal novel ecological insights at spatial scales which are germane to the management of populations and entire ecosystems. Here, we present a robust transferable deep learning pipeline to automatically locate and count large herds of migratory ungulates (wildebeest and zebra) in the Serengeti-Mara ecosystem using fine-resolution (38-50 cm) satellite imagery. The results achieve accurate detection of nearly 500,000 individuals across thousands of square kilometers and multiple habitat types, with an overall F1-score of 84.75% (Precision: 87.85%, Recall: 81.86%). This research demonstrates the capability of satellite remote sensing and machine learning techniques to automatically and accurately count very large populations of terrestrial mammals across a highly heterogeneous landscape. We also discuss the potential for satellite-derived species detections to advance basic understanding of animal behavior and ecology.

The African continent has the greatest diversity and abundance of mammals in the world[1]. This status, however, is threatened by intensive land use changes driven by increasing natural resource extraction and infrastructure development[2,3]. Even in protected areas, Africa's large mammal populations have declined by 59% in three decades[4], and many are now categorized as endangered or threatened by the International Union for Conservation of Nature. Climate change promises to only accelerate these losses, underscoring the need for advanced

monitoring techniques that can provide managers with information at a rate that keeps pace with local environmental changes[5,6].

Conventional methods for surveying large wildlife, especially in Africa, have relied on crewed aerial surveys for decades[7-11]. This approach has generated some of the longest-running ecological datasets in the world and formed the foundation of leading conservation strategies across the continent. However, crewed surveys introduce risks to human and wildlife and in many cases can only

[1]Department of Natural Resources, Faculty of Geo-Information Science and Earth Observation, University of Twente, Enschede, The Netherlands. [2]Lancaster Environment Center, Lancaster University, Lancaster, UK. [3]UK Centre for Ecology & Hydrology, Lancaster, UK. [4]School of Computing, University of Kent, Canterbury, UK. [5]Department of Ecology and Evolutionary Biology, Princeton University, Princeton, NJ, USA. [6]U.S. Army Research Laboratory, Army Research Office, Durham, NC, USA. [7]The National Academies of Sciences, Washington, D.C., USA. [8]Conservation Ecology Center, Smithsonian National Zoo and Conservation Biology Institute, Front Royal, VA, USA. [9]School of Natural Sciences, Macquarie University, Sydney, NSW, Australia. [10]Institute of Biodiversity, Animal Health, and Comparative Medicine, University of Glasgow, Glasgow, UK. [11]Geography and Environmental Science, University of Southampton, Southampton, UK. [12]Department of Ecology, Evolution and Marine Biology, University of California, Santa Barbara, CA, USA. [13]Wildlife Research and Training Institute, Naivasha, Kenya. [14]These authors contributed equally: Zijing Wu, Ce Zhang, Xiaowei Gu, Isla Duporge. ✉e-mail: t.wang@utwente.nl

provide animal counts with coarse location precision. Moreover, all crewed aerial survey techniques are subject to biases arising from detection probability, observer experience and double counting[8,12]. Uncrewed aerial vehicles (UAVs) with imaging sensors offer a promising alternative to crewed surveys in some cases[13–18]. However, like crewed flights, UAVs are generally limited by fuel or battery life and, thus, are limited in scale and can be difficult to maintain in remote locations[19]. Moreover, UAVs can disturb wildlife when flown at low altitudes[20–22], which has led to flight restrictions in some protected areas[23].

Recent advances in satellite technology have dramatically increased the feasibility of conducting uncrewed surveys in remote landscapes and at greater scales than UAVs are currently capable of. Many of the first applications of this technology focused on visualizing and analyzing easier-to-view environmental markers that, in certain contexts, provide insights to estimate population size (e.g., guano stains[24], nests[25], mounds and burrows[26]). It took less than a few years, however, for the technology to accommodate manual counts at the scale of individual animals for species in unobscured contexts (e.g., polar bears[27], albatrosses[28], and Weddell seals[29,30]). However, reliance on labor-intensive manual detection has restricted uptake by the conservation community, highlighting the need for automated techniques for processing fine-resolution satellite images.

Machine learning and the associated sub-field of deep learning, have offered promising solutions to the challenge of conducting wildlife surveys from space. Over the past decade, deep learning has been a key driver of progress in science and engineering[31]. Such advancements have had a transformative impact on the field of computer vision, where the performance of some deep learning algorithms has achieved or surpassed human-level performance in many tasks[32–36]. At the same time, new collaborations between ecologists and computer scientists have provided several key advancements in automated animal detection from satellite imagery, including detection of the world's largest marine and terrestrial vertebrates, such as whales[37] and elephants[38], using object detection algorithms. However, the performance of current object detectors suffers from the small size of the objects in imagery[39–41]. The feasibility of successfully using object detection methods is dependent on the body size of the animal: mature whales have a body length of more than 20 m[42], and African elephants are generally 3–5 m long[43], both of which have more than eight pixels along the body length axis in submeter-resolution (e.g., 0.3–0.5 m) satellite imagery.

A few studies have conducted automated surveys for smaller species with satellite images, such as for seals[44] and albatrosses[45] using pixel-based semantic segmentation algorithms. Image segmentation deep learning architectures such as U-Net[46] predict the class probability for every pixel, showing the potential to detect animals with a smaller size in satellite imagery. However, these early successes were limited to high-contrast species in homogeneous environments. The capability to reliably distinguish smaller animals (e.g., ≤9 pixels in size in satellite imagery, such as wildebeest, one of the African ungulate species) from complex backgrounds (e.g., mixed forest and savanna ecosystems) remains uninvestigated and continues to be a major question in satellite-based techniques for wildlife surveys[47].

Here, we address this shortcoming by presenting a robust framework for efficiently locating and counting wildebeest-sized animals with a body length of 1.5–2.5 m from submeter-resolution satellite imagery across a large, highly heterogeneous landscape. We do this by integrating a post-processing clustering module with a U-Net-based deep learning model, which uses high-precision pixel-based image segmentation to locate animals at the object level. We demonstrate the power of this framework by deploying it to locate and count the largest terrestrial mammal migration on the planet—the migration of white bearded wildebeest (*Connochaetes taurinus*) and plains zebra (*Equus quagga*) across the Serengeti-Mara ecosystem. Wildebeest have an estimated population of ~1.3 million individuals, making them the most numerous species in the ecosystem by an order of magnitude[48,49]. There are also over 250,000 zebras and other ungulate species that move seasonally across the system in tandem with wildebeest[48]. As a result, their annual migration drives multiple ecological processes that support the health of humans and wildlife across the region (i.e., nutrient cycling, trophic interactions, biomass removal and habitat recovery from over utilization[50–53]). In addition, the spectacle of the great migration supports a robust tourism industry, which underpins regional economies across Kenya and Tanzania. However, with the migration subject to seasonality of rainfall and habitat preference, this iconic system is facing unprecedented threats from rapid climate and environmental change[54–57]. Thus, the ability to frequently and accurately assess the status of migratory ungulate populations is key to forming conservation policies that address current threats and promote ecosystem function. In addition to supporting conservation planning in East Africa, these methodological advances stand to inform basic scientific understanding of ecological patterns and processes, such as quantitatively describing the emergent properties of animal aggregations[58,59] and answering long-standing questions about the mechanisms that drive behavioral shifts from individuals to populations. Such insights are crucial for advancing the fields of functional ecology and collective behavior, yet the technological challenges associated with studying animal aggregations in the wild have hindered scientific understanding outside of a laboratory environment[60]. Here, we take a germinal step towards overcoming such challenges by presenting a method for locating and counting large groups of animals in fine-resolution satellite imagery.

## Results
### A U-Net-based ensemble learning model for wildebeest detection

As a network designed for image segmentation tasks, U-Net allows precise pixel-level localization of a target class in an image[46]. However, it is not directly suitable for object detection applications. To address this issue, we present a U-Net-based detection pipeline that involves a post-processing module using a clustering method (Fig. 1). The pipeline is composed of three main blocks. In the first block, we subdivide the raw satellite image scenes into 336 by 336-pixel images (hereafter patches) as the input images for the model. The wildebeest in the input images are annotated as points, which are expanded to 3 by 3-pixel segments and are then converted to binary wildebeest/non-wildebeest image segmentation masks. In the second block, the satellite image patches and the corresponding masks of labeled wildebeest are fed into the U-Net model, which predicts the probability of wildebeest presence for each pixel. The U-Net model has a U-shaped symmetrical encoder-decoder structure that consists of a contracting path on the left, which extracts high-level features, an expanding path on the right that increases the resolution, and multiple levels of skip connections between two paths that allows for precise localization. To increase the robustness of the model, we adopt ensemble learning through a *K*-fold splitting method. The training dataset is split into ten folds, with nine folds used for training and the remaining fold used for validation. This ensemble block introduces variation in the training and validation datasets and achieves 10 individual base models. We then summarize the predictions by averaging the probability maps produced by these 10 base models. In the last post-processing block, we convert the pixel-wise prediction into wildebeest individuals through *K*-means clustering. The clumped wildebeest pixels were disaggregated by *K*-means clustering to separate individual wildebeest (Supplementary Fig. 1), which were used as the final outputs for evaluation at the individual level. Note that as wildebeest is the dominant ungulate species in the system and most animals we located and counted were wildebeest, we refer hereafter to the migratory ungulates detected by our model as wildebeest for the purpose of simplicity.

We applied the pipeline to satellite images acquired over six years (August 2009, September 2010, August 2013, July 2015, August 2018, and October 2020) covering 2747 km² in the Serengeti-Mara ecosystem (Fig. 2). The images were captured by different satellite sensors with distinct spatial resolutions ranging from 38 cm to 50 cm, including GeoEye-1 (GE01), WorldView-2 (WV02) and WorldView-3 (WV03). Each individual wildebeest in the satellite imagery was represented by ~3-to-4 pixels in length and 1-to-3 pixels in width, with 1 or 2 relatively darker pixels in the center, including the shadow of the body (Fig. 3). The training dataset contained 1097 image patches captured from these six years, including 53,906 manually labeled wildebeest points across various environmental conditions. We incorporated labels created by four independent expert observers by majority voting. The details about the level of their agreement are presented in Supplementary Table 1. During the labeling process, we used a set of reference satellite images acquired on different dates, but with the same background landscapes for cross-referencing to ensure the labels were moving animals and were not similar-looking static objects (e.g., termite mounds, small bushes). The acquisition dates and spatial resolutions of the reference images are presented in Supplementary Data 1. During model training, the training dataset was split randomly into 10 folds, among which nine folds were used for training and the remaining one fold was used for validation.

To evaluate model performance, we used a stratified random sampling method to select test sample plots across the images in each year to ensure their representativeness and independence from the training dataset. The strata are based on the number of animals in the image patches. The distribution of the number of animals per image is

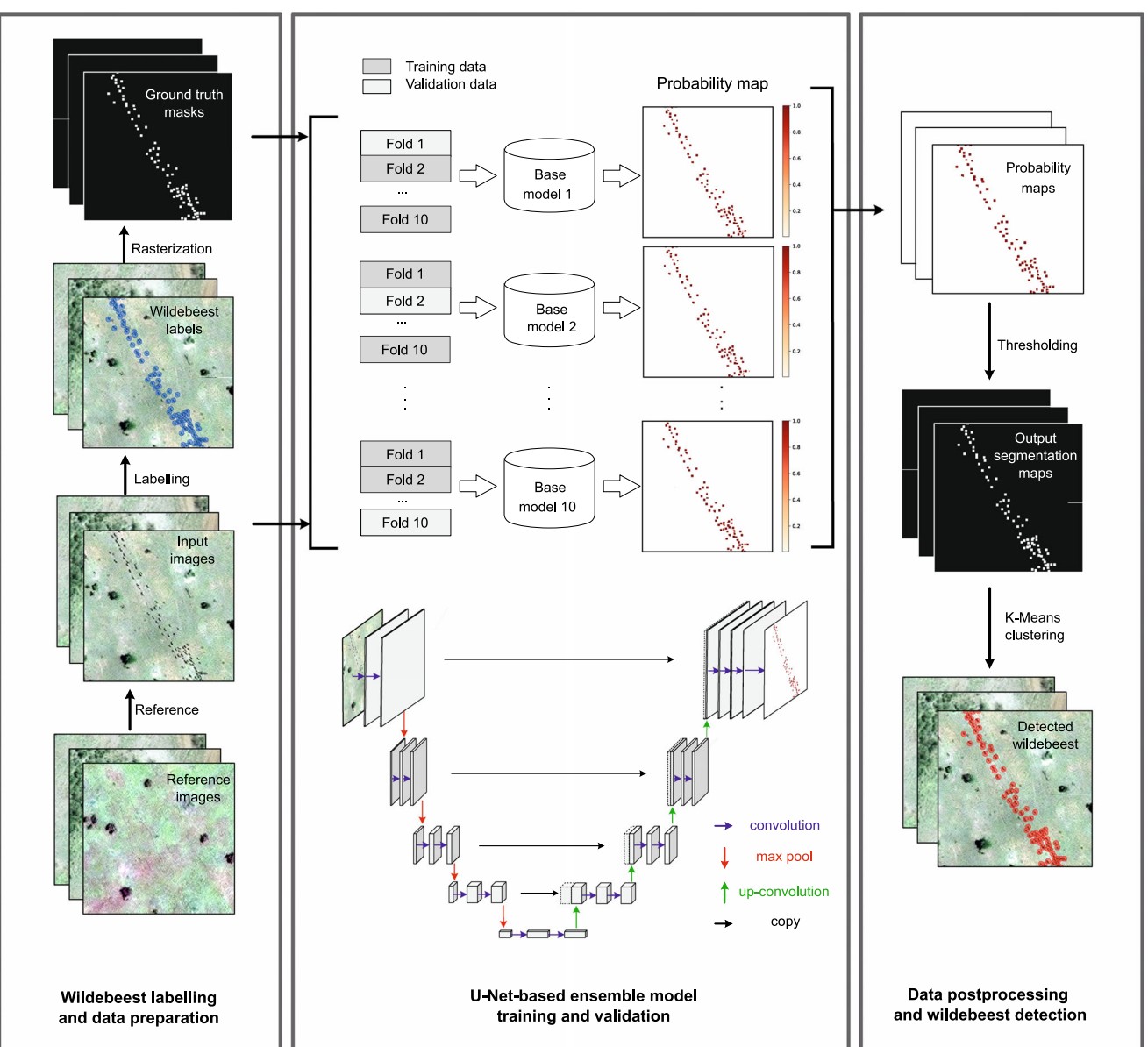

**Fig. 1 | Model framework.** The wildebeest detection pipeline consists of three main blocks: 1) The wildebeest are labeled in the satellite imagery and the masks are generated; 2) The satellite images and the masks are fed into the U-Net-based ensemble model for model training/validation and to produce the wildebeest probability maps; 3) The probability maps produced by the 10 base models are averaged to obtain the final predictions and the wildebeest individuals are detected using *K*-means clustering. The blue dots on example image of wildebeest labels represent manually annotated wildebeest labels. The red dots on example image of detected wildebeest represent wildebeest detected by the framework. In the U-Net architecture visualization, each box in gray color represents a multi-channel feature map layer. The gray box with dashed line represents copied feature map from the left part. Each arrow represents an operation. Satellite image © 2010 Maxar Technologies.

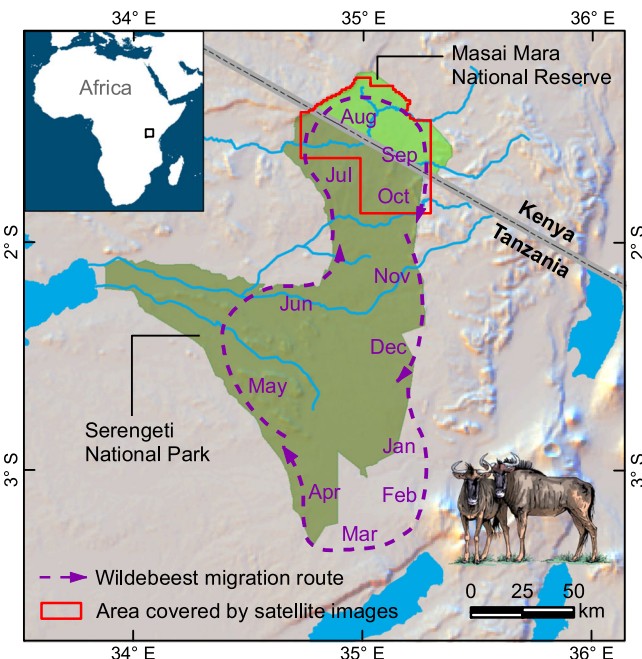

**Fig. 2 | Study area map.** The satellite imagery used in this research cover mainly the Masai Mara National Reserve and the northernmost section of the Serengeti National Park (the area outlined in red). The wildebeest typically migrate over 1500 km on average every year (the purple dashed line). During June and August, the wildebeest migrate from the Serengeti plains in Tanzania into the Masai Mara National Reserve and then spread to the east crossing the Mara River in September. Then during November and December, they move south to the southern Serengeti. Image credit: EreborMountain/Shutterstock.com for the wildebeest art photo.

summarized in Supplementary Fig. 2. In total, we selected 2700 test images containing 11,594 wildebeest individuals. Key information about the images used and the size of training and test dataset is summarized in Supplementary Table 2. More details about the sampling method and data preparation process are described in the Methods section. We calculated the model performance for each year and also calculated the overall accuracy by combining all the test datasets. The accuracy (precision, recall, F1-score) was evaluated on a per-individual basis as demonstrated in Fig. 4. The model achieved an overall F1-score of 84.75% with a precision of 87.85% and a recall of 81.86%. The model performed well in each year (Supplementary Table 3): all F1-scores were above 80% (between 80.40% and 91.70%). The precision across the six years varied between 82.68% and 97.80% and recall between 74.00% and 87.52% (Fig. 5a). This indicates that the model has good generalization ability across varied image resolution (from 38 to 50 cm), despite the great temporal and spatial variation in landscape type, ecological conditions, and mode of image acquisition over different years.

To validate the advantage of using an ensemble model, we also compared the performance of the ensemble model with the individual base models. The original training dataset was split into 10 folds, nine of which were used for training and the remaining fold for validation, resulting in 10 models trained on various datasets. The predictions of the 10 models were averaged to obtain the final results. We assessed the performance of each individual model using the Precision-Recall curve and Area Under the Curve (AUC). The ensemble model achieved an AUC of 0.88, which is significantly higher than all other base models (Fig. 5b). We also compared the F1-score: the F1-score of 10 base models on average is 78.22% (±0.86%), also lower than the F1-score of ensemble model (84.75%). A more detailed comparison is listed in Supplementary Table 4.

## Model transferability

To assess the temporal and spatial transferability of the model, we ran two tests:

1. Transferability of the model to a temporally different dataset: we selected the image from 2015 as an independent test dataset and trained the model with wildebeest labels from the other five years (2009, 2010, 2013, 2018, 2020). The 2015 dataset was an unseen image captured with a different sensor, with the finest spatial resolution (38 cm of WV03 versus 42–50 cm of GE01 and WV02). The model achieved high accuracy on this new dataset, with a precision of 90.77%, recall of 95.61%, and F1-score of 93.13%. Such high accuracy indicates the model can be transferred to a temporally different dataset without adding additional training samples and still demonstrate excellent performance.

2. Transferability of the model to a spatially different dataset: we selected the images from 2020 as an independent test dataset and trained the model with wildebeest labels from the other five years (2009, 2010, 2013, 2015, 2018). The coverage of the 2020 data is on the east side of Masai Mara National Reserve and Serengeti National Park, which is outside the coverage of the remaining datasets, and its spatial resolution is the coarsest (50 cm of WV02) of all years. The model achieved a 96.98% precision, showing that the model is able to avoid false positives without adding any new training samples for this new task with different landscapes and ecological conditions. The recall score is 60.65% (with F1-score of 74.63%), indicating the ability to detect all positives can still be improved by adding more samples from the 2020 dataset.

## Wildebeest detection and counting

To detect and count migratory wildebeest within the area, we applied the U-Net-based ensemble model trained with full training datasets from all six years to the entire satellite imagery dataset that covered a large portion of the dry-season range of migratory wildebeest. Figure 6 shows examples of the detection across varied landscape characteristics including savanna, woodland and riverine forests. The detection results demonstrate the model's robustness to variation in three dimensions: (1) variation between different satellite sensors, namely, various spatial resolutions over the six different years; (2) variation in the landscape context, such as river, woodland, bushland and grassland, with the risk of confusion with background objects such as termite mounds, small bushes and shadows caused by terrain, and (3) variation in the wildebeest aggregation patterns, such as scattered, linear and clustered. Further examples of detected wildebeest patterns across very large areas can be found in Supplementary Fig. 3-8 and Supplementary Data 2. The method resulted in a sum count of 480,362 (ranging between 470,121 and 490,603) individual wildebeest (F1-score: 84.75 ± 0.18%) across the whole dataset (Table 1). See Fig. 7 for the location and coverage of the imagery of each year and Table 1 for the number of animals detected in each year.

To further analyze the spatial distribution pattern of the migrating wildebeest in the Serengeti-Mara ecosystem, we calculated the wildebeest count per km$^2$ in each scene and plotted the resulting histogram (see Fig. 7a–f). The maximum wildebeest density displays great variation across months in the dry season (July–October). Peaks in wildebeest density appear in August in the western Masai Mara National Reserve (more than 4000 to 6000 individual wildebeest per km$^2$). In September, the peak wildebeest density is ~3000 per km$^2$, while in July and October, the maximum density is between 1500 and 2000 per km$^2$. The spatially and temporally varied density is visualized in the hotspot maps in Fig. 7.

We also present the enlarged hotspot map in Fig. 8. The high densities and dense clusters of wildebeest were observed in the three representative images from August (2009, 2013, 2018). Variation in this pattern is evident in the lower wildebeest densities observed in the representative image analyzed from September 2010 and the more

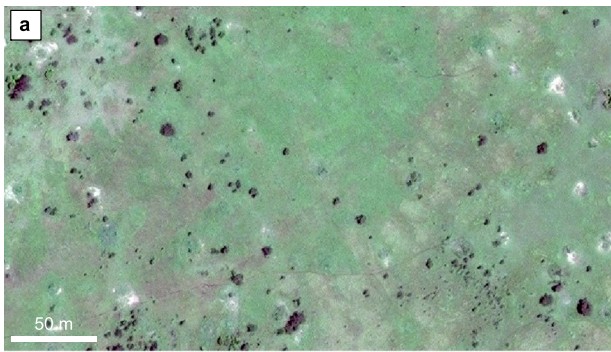

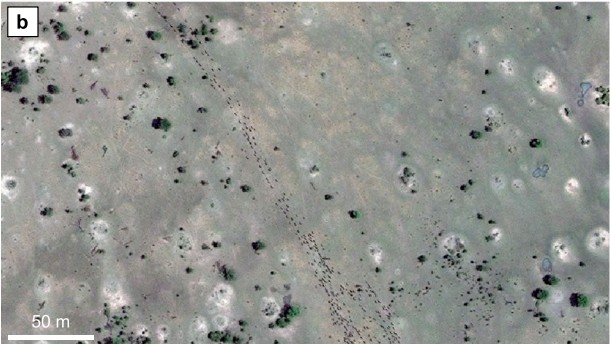

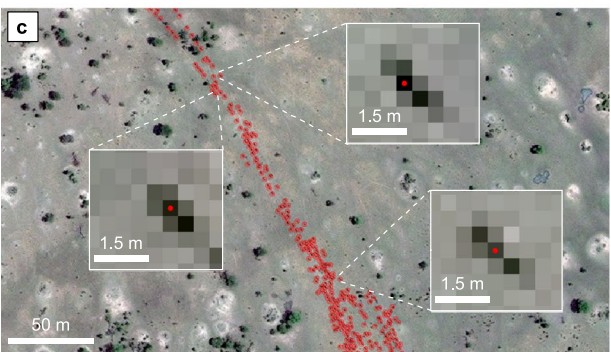

**Fig. 3 | Labeling the wildebeest on the satellite image. a** The reference satellite image that was used for cross-referencing while labeling the wildebeest. This example image was acquired on May 17th,2012. **b** The satellite image acquired on September 24th, 2010 for wildebeest labeling. **c** Wildebeest labels on B. The red points denote wildebeest annotations. The zoomed boxes are three examples of the wildebeest labels on the GE01 image with 44-cm resolution. Satellite image © 2010 Maxar Technologies.

scattered distribution observed spread out over a larger area in the October 2020 image. The distribution dynamics observed comply with the general wildebeest migration patterns shown in Fig. 2. The wildebeest migrate to the north towards the Mara Triangle in July and August, and aggregate there for grazing before moving further southeast across the Masai Mara National Reserve in September, and spread south into the vast Serengeti National Park in October, as shown in the sparse distribution in the hotspot map.

## Discussion

The detection pipeline presented here demonstrates the potential for deep learning techniques to efficiently track fine-scale environmental changes through automated, satellite-based wildlife surveys. To create outputs that would have real-world utility to researchers and managers, we deployed our model at an especially large spatial scale (2747 km²) and validated it on a dataset that varied in space, time, and resolution. This approach yielded highly accurate results (with an overall F1-score of 84.75%) and the largest training dataset ever

published from a satellite-based wildlife survey (53,906 annotations). In addition to its size, the landscape diversity captured by this dataset will facilitate model transferability to applications in similar environmental contexts, such as future satellite-based wildebeest census surveys at the ecosystem scale. Although generalization of our model is inherently limited to wildebeest-like animals in open landscapes, the pipeline itself is generic and can be applied to other animal detection applications after retraining.

Beyond providing a truly open-source and transferable method for satellite-based wildlife surveys, our approach holds extreme promise for scaling spatially to produce the first ever total counts of migratory ungulates in open landscapes. Such information is particularly important to the management of aggregating species like wildebeest because their heterogeneous and autocorrelated grouping patterns violate the assumptions of most statistical methods for estimating population abundance from survey data[61]. As a result, traditional methods are prone to systematic undercounts and high uncertainty[61]. An automated total count would eliminate the need for statistical inference and potentially produce a correction factor that could be used to reduce error in historic estimates through post-hoc analysis. While a total count would still assume near-perfect detection of animals, we note that this ideal may be achieved in open systems where biological cycles drive predictable periods of aggregation. For example, wildebeest could be censused while gathered to calve on the nutritious shortgrass plains of Serengeti, caribou could be censused while gathering to cross seasonal ice floes in the arctic, and white-eared kob could be imaged while concentrated in low-lying meadows along the margins of major watercourses during the dry season.

A next valuable step in the science of enumerating large mammal populations using the proposed satellite-based method will be ground-truthing the predictions against both historical and contemporary estimates of population size derived using traditional methods (e.g., ground-based or aerial counts). For the present case of the wildebeest population, satellite-derived counts should be compared against the data collected every 2–3 years using aircraft surveys in the Serengeti National Park[7,62]. Comparisons can be conducted both at the transect level (with satellite image acquisition synced to the timing of aircraft transects—although noting that temporal alignment of surveys with suitable conditions for both survey types can be challenging) and at the whole population level via data extrapolation.

In addition to facilitating total counts for multiple species, the ability to observe expansive herds of migratory ungulates from space presents an exciting opportunity for the study of the ecology of animal aggregations from an entirely novel perspective. For example, the spatially explicit point data produced by our model can be readily analyzed as an ecological point process[63] to facilitate the first-ever quantitative descriptions of wildebeest herding patterns in the wild. Such insights are crucial for answering key ecological questions about social and environmental drivers of animal behavior and identifying emergent biological patterns that scale from individuals to populations[63]. Likewise, a robust time series of satellite images may be used to extend previous work on the ecology of large-scale aggregation patterns of wildebeest across the landscape[64]. We demonstrate the potential for our pipeline to inform this approach by producing density plots from model outputs, which can then be mapped and analyzed within their native environmental context (Fig. 8). This ability to track the distribution of large animal aggregations over time is important for guiding adaptive management of mobile species and for deriving a systematic understanding of population-level responses to rapid environmental change.

Another potentially promising application of the proposed method would be the detection of large mammal migrations that have not previously been documented. Despite the charisma of such fauna, the migrations can go uncharacterized and are infrequently discovered or rediscovered (e.g., the Burchell's zebra migration in

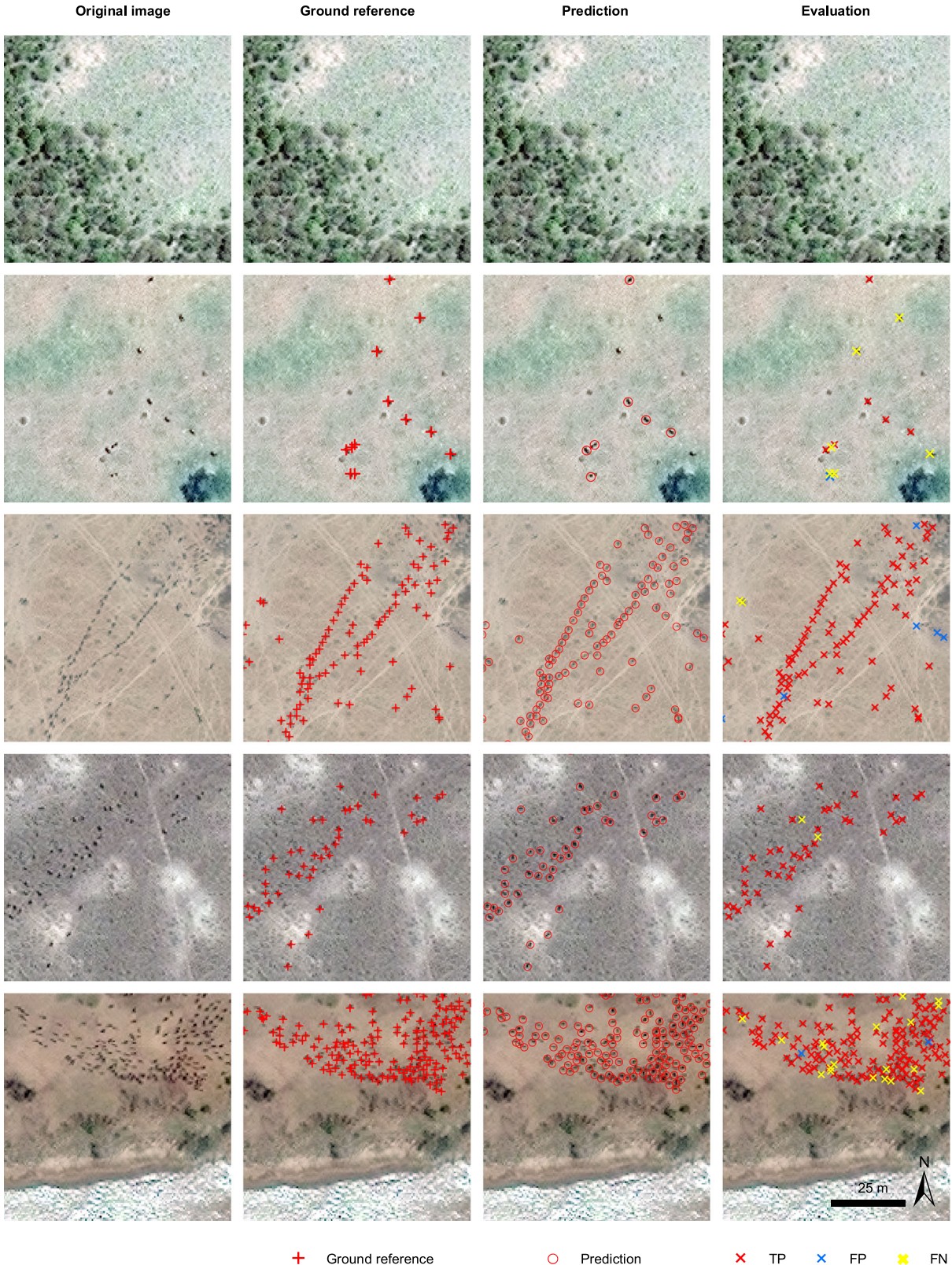

Original image | Ground reference | Prediction | Evaluation

+ Ground reference ◯ Prediction ✕ TP ✕ FP ✕ FN

**Fig. 4 | Examples of model evaluation on individual wildebeest.** In the Evaluation column, the predictions that match the ground references are True Positives (TP, red crosses), and those that do not match are False Positives (FP, blue crosses). Ground references that were not detected by the model are False Negatives (FN, yellow crosses). The examples are taken from the test set of 2009–2020, showing that the model avoids most of the background objects that have similar size and color to wildebeest objects, such as small bushes, shadows on the edges of ponds, and roads. Satellite image © 2009–2020 Maxar Technologies.

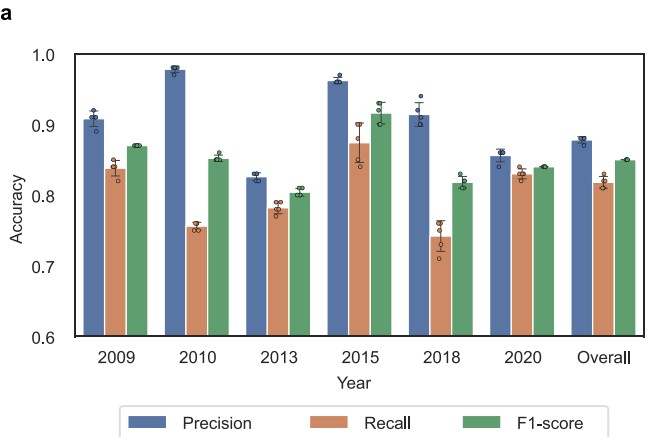
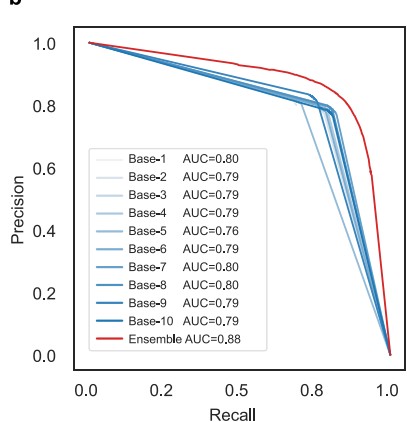

**Fig. 5 | Model performance. a** The wildebeest detection accuracy of the U-Net-based ensemble model for each of the six years and the whole dataset. Error bars represent mean values ± SD (*n* = 5). **b** The Precision-Recall curve of the ensemble model and each base model. The red line (representing the ensemble model) lies above all other blue curves (representing the individual base models), indicating greater accuracy.

Nambia/Botswana;[65] white-eared kob in South Sudan[66]). Given the advantages of surveying at large scales, satellite imaging techniques, coupled with GPS tracking of individual animals, could provide a powerful methodological combination for detecting or confirming such migrations. GPS tracking data could benefit the survey by giving prior information about the potential range, while regularly acquired satellite imagery can be used to identify the migration routes of large animal groups over time, as satellite imaging at high time frequency becomes possible. Such methods are also especially useful for detecting and studying wildlife migrations in remote or insecure regions[66].

Despite the clear potential for satellite-based wildlife surveys to advance both basic and applied research, this technology is still limited by the inherent challenge of distinguishing small objects from only a few pixels on satellite imagery. While the commonly used deep-learning based object detectors for animal detection are confined by the size of the object on the image[37,67,68], our method addresses this challenge by utilizing a class of convolutional neural networks (specifically the U-Net model) designed for pixel-level segmentation, thus enabling detection of objects that occupy less than 9 pixels. This method uses ensemble learning to further increase the accuracy of individual U-Net models. By combining the clustering module, the ensemble model can separate multiple clustered animals and identify individual animals with high accuracy and efficiency. This is an advancement compared to previous studies, which had lower detection accuracy for similarly sized animals (e.g., seal detection with <50% accuracy[44]), or focused on identifying large animals in homogeneous environments (e.g., whales[37]).

Nevertheless, the current limitation of satellite image resolution impacted our study by preventing distinction between wildebeest and other species of similar size, including domestic cattle (*Bos taurus*), topi (*Damaliscus korrigum*), Coke's hartebeest (*Alcelaphus buselaphus cokii*), and eland (*Taurotragus oryx*). While we controlled for the most numerous species (e.g., cattle) by limiting collections to sites and seasons with minimal overlap, finer-resolution imagery (for example, <10 cm) will be required to discriminate these species. We also note that smaller-bodied species (e.g., gazelle) were not visible at the current resolution, but larger species (e.g., hippos and elephants) were successfully excluded by the model. Given these promising results, we are confident that pending technology will rise to meet the demand to resolve smaller species, as multiple satellite companies have already announced the arrival of breakthrough technologies that will make sub-daily, sub-50 cm imaging a reality. One limitation in satellite imaging wildlife currently is the cost of very-fine-resolution imagery.

However, costs are falling as more companies are now offering sub-meter imaging capabilities from multiple constellations at lower prices. In addition, many satellite providers (e.g., Maxar, Airbus and Planet) are providing more opportunities for researchers to access sub-meter imagery at low or zero cost.

As more fine-resolution constellations come online, we anticipate that satellite-based wildlife surveys will become increasingly affordable and accessible. We aim to capitalize on this technological moment by validating a data pipeline, which advances the scale and scope of current techniques to include medium-sized mammals in highly heterogeneous landscapes. While there are many applications for this pipeline, we wanted to demonstrate its potential to monitor animals across an area of unprecedented size by counting hundreds of thousands of wildebeest in the Serengeti-Mara ecosystem. When combined with anticipated advances in satellite imaging, the outputs of our model will improve the frequency and accuracy of population estimates for multiple species in open landscapes and produce novel datasets for investigations of animal behavior, ecosystem ecology, and global change biology.

## Methods
### Satellite imagery
The satellite imagery used for wildebeest detection and counting includes nine multispectral images captured by three satellite sensors (GeoEye-1, WorldView-2, and WorldView-3) over six years in the Serengeti-Mara ecosystem. We selected these images from the archived very-fine-resolution satellite images acquired by the Maxar Worldview constellation, which can cover more than 3.8 million square kilometers per day and has a revisit rate of 1-2 times per day. The images we used mainly cover the Masai Mara National Reserve and the northernmost section of the Serengeti National Park (see Fig. 2 of the study area). The images cover 2747 km² within the delimited boundary. The spatial resolution varies from 38 to 50 cm (see Supplementary Table 2 of image resolution and date). Most of the acquired images were delivered as pan-sharpened products, while the WorldView-2 images in 2020 were pan-sharpened using the UNB-pansharp method[69]. The pre-processed satellite images have four bands: Red, Green, Blue and Near-Infrared. All the images are covered by cloud by less than 2%. In addition, another set of eight satellite images covering the same area as the images above, but acquired on different dates are used as a set of reference images for wildebeest labeling. Details of the input satellite images and the reference images are listed in Supplementary Data 1.

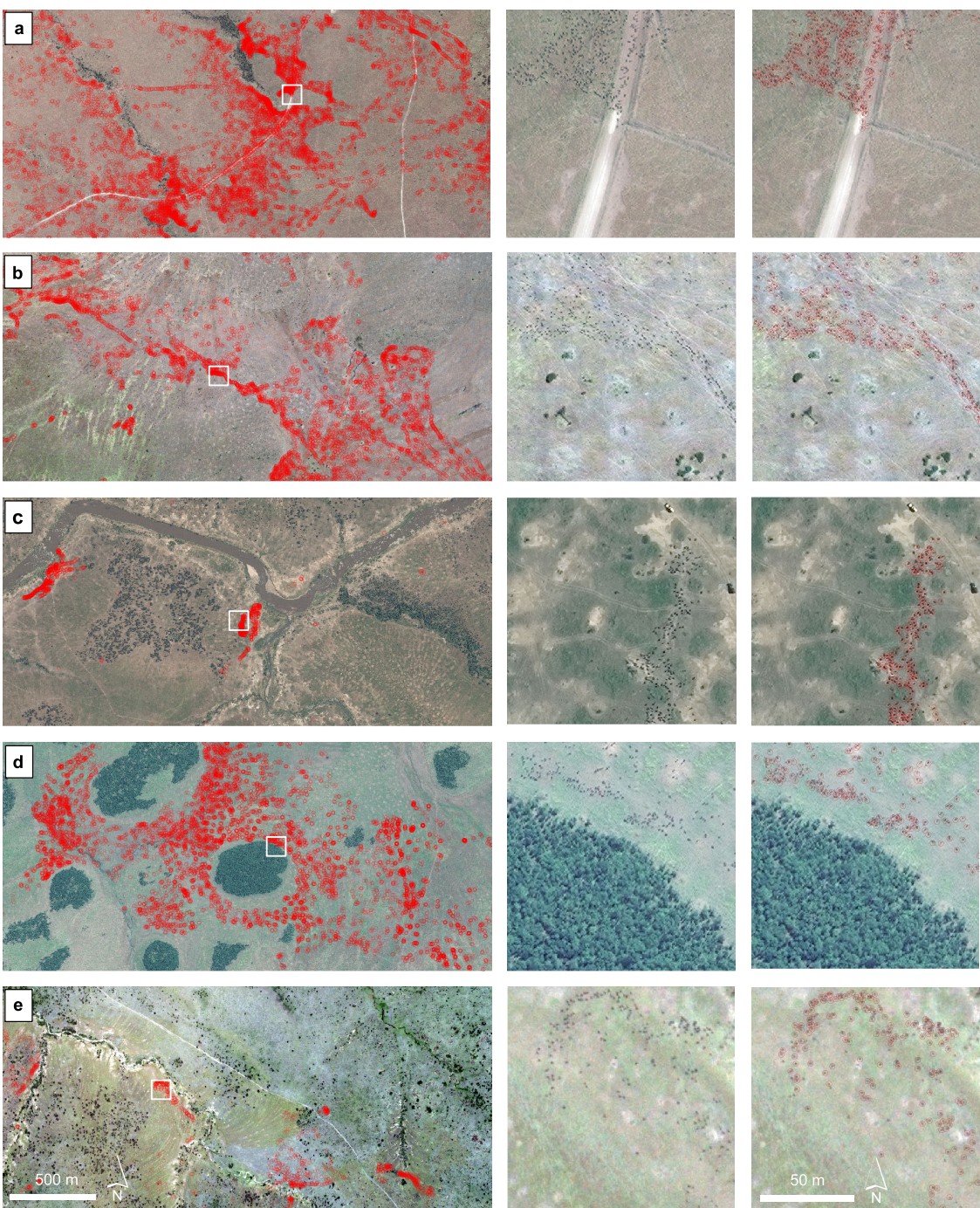

**Fig. 6 | Detecting wildebeest across different landscapes with variation in wildebeest spatial clustering patterns.** The figures in the first column show the detected wildebeest (red circles). The second column is a zoom of the imagery covered by the white square in the first column. **a** Detected wildebeest in GeoEye-1 imagery acquired on August 11th, 2009. In the zoomed-in image, the wildebeest are crossing the road near a dry riverbed. **b** Detected wildebeest in GeoEye-1 imagery acquired on August 10th, 2013. Wildebeest herd in open grasslands. **c** Detected wildebeest in WorldView-3 imagery acquired on July 17th, 2015. The wildebeest prepare to cross the Mara River. **d** Detected wildebeest in GeoEye-1 imagery acquired on August 2, 2018. Herds of wildebeest avoid the closed woodlands. **e** Detected wildebeest in WorldView-2 imagery acquired on October 8th, 2020. The wildebeest herds move through open woodlands and grasslands. These examples also show the heterogeneity between the satellite images, inclusive of spectral variation and different levels of contrast between the wildebeest and the background. Satellite image © 2009–2020 Maxar Technologies.

## Labeling the wildebeest

In the satellite imagery, we labeled the individual wildebeest as points in vector format. On the true color composite image, a wildebeest is a group of gray-brownish pixels with a dark black pixel commonly in the center representing the animal's neck and spine with a black mane. Each wildebeest individual in the image was about 3 to 4 pixels in length and 1 to 3 pixels in width, with 1 or 2 relatively darker pixels in the center as shown in Fig. 3. Therefore, for each wildebeest, we labeled one point at the center of this wildebeest segment, and then expanded the point to a polygon with a size of 3 by 3 pixels, such that the polygon covers most of the wildebeest pixels. The wildebeest labels were derived using majority voting from visual interpretation

**Table 1 | The number of wildebeest detected and counted in six different years of satellite imagery**

| Date | Number of wildebeest (At 95% confidence level, $n = 5$) |
|------|---------------------------------------------------------|
| 11/Aug/2009 | 122,750 ± 1905 |
| 24/Sep/2010 | 79,039 ± 782 |
| 10/Aug/2013 | 149,232 ± 6623 |
| 17/Jul/2015 | 15,855 ± 672 |
| 02/Aug/2018 | 44,832 ± 3177 |
| 08/Oct/2020 | 68,655 ± 1103 |

undertaken by four expert observers of the same satellite image, cross-referenced against another (reference) satellite image acquired in a different year. The purpose of using reference images was to distinguish between wildebeest and spectrally similar background objects, such as small bushes and the shadows of termite mounds, which are static in both images.

## Training and test dataset

For each satellite image, we built a grid system with a cell size ranging from 150 m to 170 m, dependent on image resolution. Each grid covered 336 × 336 pixels, which was the size of the image patch for model training. The training and test datasets were sampled based on the cell units of the grid. In the training dataset, we selected a total of 1097 training grids, covering different types of landscapes and various wildebeest abundances across all six years. The training dataset contains 53,906 wildebeest, occupying 27.13 km$^2$, which is 0.7% of the whole area. The test datasets were sampled using the proportionate stratified random sampling method on each image date, containing 2700 sample grids with 11,594 wildebeest. We adopted this method to guarantee the representativeness of the test dataset.

The strata of the test dataset were based on the wildebeest density in the grids in accordance to the spatially imbalanced distribution of wildebeest, ensuring the test dataset contains sample grids with different levels of animal density. Therefore, preliminary information on wildebeest density was required. We first built an initial test dataset using a random sampling method and trained a model to achieve an acceptable detection performance on the initial test dataset. Then we applied the preliminary model to the whole imagery dataset to detect and count the wildebeest, which were used to estimate the wildebeest density in all the grid cells. The grid-level wildebeest density was used as the criteria to classify the grid cells into one of four categories (low density, medium density, high density and very high density) based on the mean and standard deviations. Supplementary Fig. 9 shows an example of the wildebeest density map in the year 2009 for sampling. Majority of the grids have low density of animals. We determined the test sample size as 100 or 200 test grid cells depending on the area covered by each image, and then selected a proportionate number of samples randomly within each category to build the final test dataset. For example, as there was a single image collected on 10 August 2009, 100 test samples were selected from it. Since there are two images on 13 August 2013, 200 test samples were chosen from them. For images collected on 08 October 2020, the area was much larger and the wildebeest density was rather low. As a result, we selected 1900 image grid cells for testing. The sample size for the year 2020 was relatively large to ensure the test datasets covered sufficient wildebeest-abundant image patches. In total, there were 2700 test grids for all six years, occupying 1.7% of the entire dataset. We manually labeled all the wildebeest in the test sample grids.

## Training the U-Net based ensemble model for wildebeest detection

Before incorporating the training dataset into the model, we first pre-processed the images and labeled wildebeest to fit the requirements of

the input data. The wildebeest polygon labels were rasterized into a small patch with 3 × 3 pixels to represent the wildebeest segments. The segments were then used to generate the binary masks, including the wildebeest pixels and non-wildebeest pixels. The masks have the same size as the corresponding satellite sensor gridded images. The gridded images and the binary masks were cropped into patches with 336 × 336 pixels. Then all data patches were augmented using horizontal flip, vertical flip, and 90° rotation to increase sample variation. These data augmentation techniques can help prevent overfitting and increase the generalization capability of the model on unseen data with unfamiliar patterns[70]. All the training image patches and the masks from the six different years were combined to train the U-Net deep learning model.

The U-Net architecture is a type of convolutional neural network designed originally for biomedical image segmentation[46], which has subsequently been applied widely in other applications, including remote sensing image segmentation. U-Net uses a U-shaped symmetrical encoder-decoder structure that consists of a contracting path on the left and an expanding path on the right[46] (Fig. 1). The contracting path encodes high-level contextual features through successive layers, which generates low-resolution, but high-dimensional feature maps. The expanding path decodes the information of these feature maps and up-samples the image to obtain the original resolution step-by-step. The up-sampled output is concatenated through skip connections with the corresponding feature map (with the same spatial resolution) in the contracting path on the left, thus, merging both sources of information to provide evidence for classification, and to support precise localization of the obtained semantic information. The last layer of the model maps the feature maps into the class number for each pixel in the original image using a sigmoid activation function, resulting in a probability map with a value ranging from 0 to 1 representing the wildebeest presence probability as the final output of the U-Net model.

We employed the ensemble learning approach[71–73] to increase the generalization capability and robustness of the U-Net model. We split the training dataset into $K$ folds ($K = 10$ in this research), of which $K-1$ folds were used for training the U-Net model, and the remaining one was used for validation. Therefore, a total of $K$ individual U-Net models were trained and validated with different subsets of the data. Then the $K$ models were combined to construct the final ensemble model, where the probability predictions of the base models were first normalized to the scale of 0 to 1 using the standard min-max approach and then averaged to produce the final outputs as depicted in Fig. 1.

To address the imbalance between the wildebeest and non-wildebeest classes, we adopted a weighted loss function, namely, the Tversky loss function[74], to measure the discrepancy between the predictions and ground references. The parameters of the Tversky loss, $\alpha$ and $\beta$, are the penalty weights for False Negatives (FN) and False Positives (FP), respectively, and the sum of $\alpha$ and $\beta$ is 1 (Supplementary Equation (1)). Considering that wildebeest detection from satellite images is a highly imbalanced problem, namely, the percentage of wildebeest pixels is less than 1% in the training imagery, the model tends to predict all the pixels into non-wildebeest pixels to achieve high overall accuracy. By increasing $\beta$, emphasis is added to minimize the number of misclassified wildebeest pixels. The parameter $\beta$ was finely tuned over a range of values (0.01, 0.1, 0.2, 0.3, 0.4, 0.5, 0.6, 0.7, 0.8, 0.9, 0.99) to reach the optimal trade-off between FPs and FNs. We used the dataset of 2009 in a sensitivity analysis to evaluate how different settings of $\beta$ influence the model performance and the optimal parameters used were $\alpha = 0.1$ and $\beta = 0.9$ (Supplementary Table 5).

The model was trained with the Adam optimizer using an initial learning rate of 0. 0001[75]. The learning rate was reduced by a factor of 0.33 when the loss on the validation set stopped improving after 20 epochs. The weights in the convolution layers were initialized by the He_normal kernel initializer[36]. The dropout rate[76] was set to 0 as

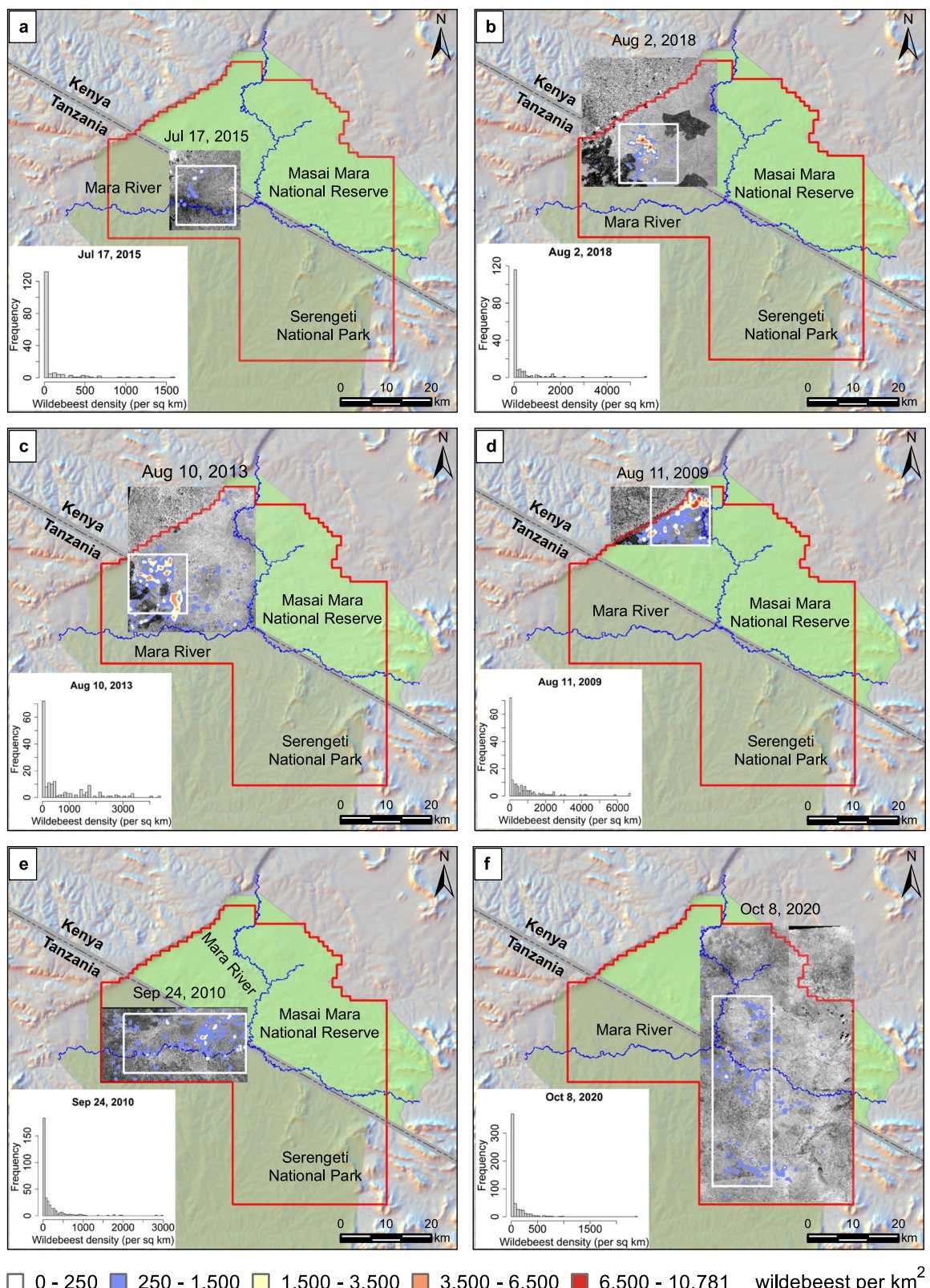

0 - 250   250 - 1,500   1,500 - 3,500   3,500 - 6,500   6,500 - 10,781   wildebeest per km$^2$

preliminary experiments showed that a higher dropout rate did not increase significantly the model performance. The batch size was 12, and the model was trained for 120 epochs. The model generating the smallest loss on the validation dataset amongst all epochs was selected as the final model. The software was implemented using TensorFlow[77] 2.1.0, and Python 3.7. The model was trained on Azure Virtual Machine with NVIDIA Tesla V100 GPU supported by Microsoft AI for Earth.

We post-processed the outputs of the ensemble model to obtain precise wildebeest point predictions. The outputs of the base U-Net models were probability maps of wildebeest presence. The probability map of each base model was first rescaled into the range of 0 to 1 (if the maximum value is greater than 0.05) and then averaged to obtain the final probability map as the output of the ensemble model. Each pixel on the final probability map was then classified as either wildebeest or

**Fig. 7 | Spatial distribution of detected wildebeest from July to October in 2009-2020.** The area outlined in red represents the study area, covering the Masai Mara National Reserve and the northernmost section of the Serengeti National Park. The area outlined in white indicates the corresponding area presented in Fig. 8. The histogram shows the calculated wildebeest frequency distribution for each scene. **a** Spatial distribution hotspot map of wildebeest detected in July 2015. The image is located in the northernmost section of Serengeti National Park with the Mara River flowing through. The maximum wildebeest density is about 1500 per km². **b** Spatial distribution hotspot map of wildebeest detected in August 2018. The image is located in the Mara Triangle inside the Masai Mara National Reserve, covering the border of Kenya and Tanzania. The wildebeest are near the border and the density peak is more than 4000 individuals per km². **c** Spatial distribution hotspot map of wildebeest detected in August 2013. The image covers the Mara Triangle in the Masai Mara National Reserve and the northern section of the Serengeti National Park. The wildebeest are mostly distributed in the Serengeti National Park near the border and the density peak is about 4000 individuals per km². **d** Spatial distribution hotspot map of wildebeest detected in August 2009. The image is located in the northwest corner of the Masai Mara National Reserve. The wildebeest density peak is about 6000 individuals per km². **e** Spatial distribution hotspot map of wildebeest detected in September 2010. The image is located in the north Serengeti National Park with the Mara River flowing through. The wildebeest are mostly on the north side of the Mara River and the density peak is about 3000 per km². **f** Spatial distribution hotspot map of wildebeest detected in October 2020. The images cover the east side of the Mara National Reserve and northeast Serengeti National Park. The wildebeest span sparsely across the Mara National Reserve and Serengeti National Park and the density peak is about 2000 per km². The maximum wildebeest density displays a large difference in terms of months in the dry season. Satellite image © 2009–2020 Maxar Technologies.

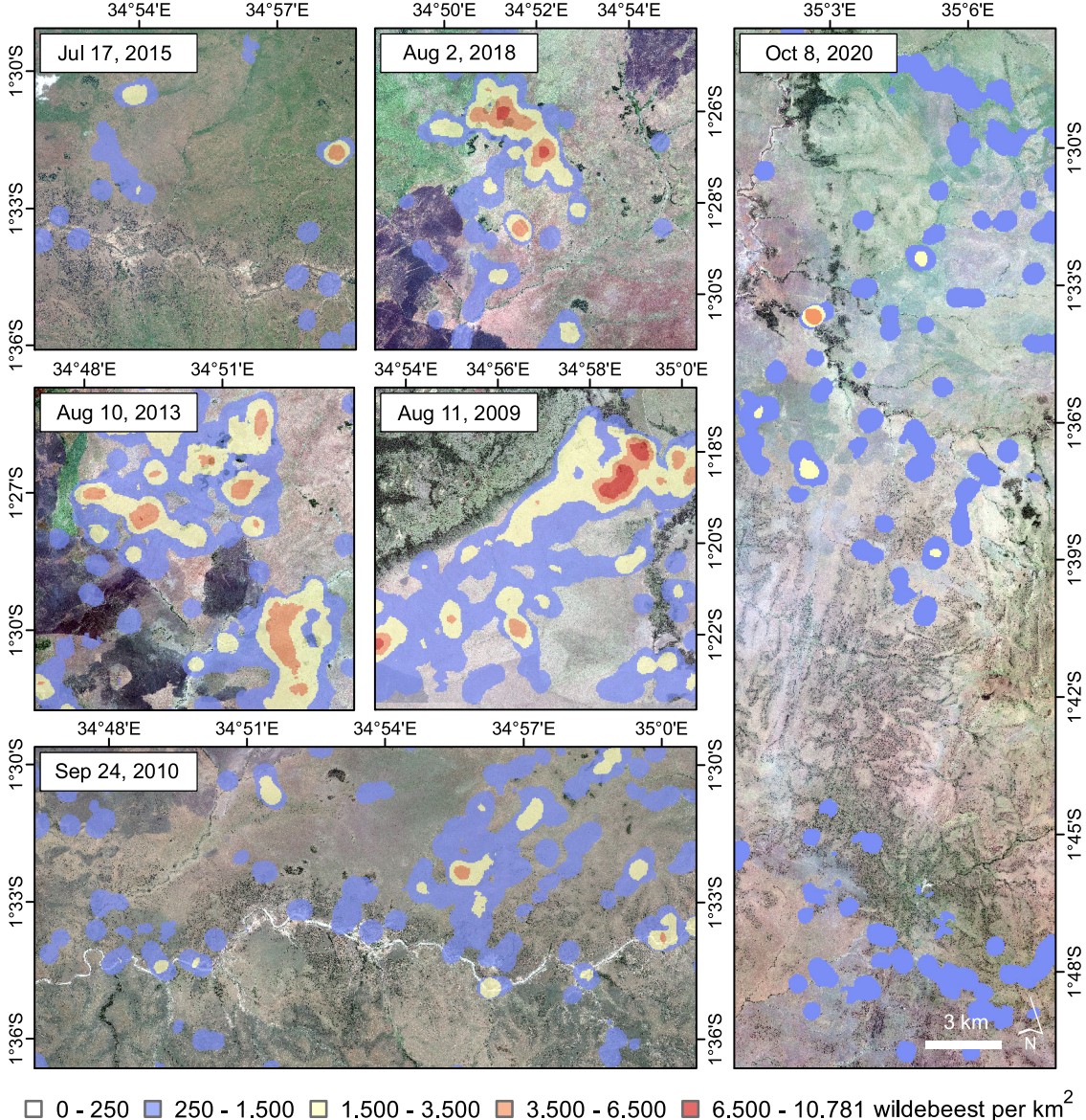

☐ 0 - 250   ☐ 250 - 1.500   ☐ 1.500 - 3.500   ☐ 3.500 - 6.500   ☐ 6.500 - 10.781 wildebeest per km²

**Fig. 8 | Hotspot map and spatial density of wildebeest over time (from July to October, 2009 to 2020).** In this figure, a subset of each timeframe was taken for display purposes and the hotspot map was produced for each timeframe with a cell size of 100 m and a radius of 500 m using Point Density tool in ArcGIS. The density of wildebeest varies from 0 to more than 10,000 wildebeest per km², and it shows a clear spatial variation of wildebeest aggregation patterns in different months. Satellite image © 2009–2020 Maxar Technologies.

non-wildebeest using a threshold of 0.5 (Supplementary Fig. 10). We converted the raster results of wildebeest segments into points that represent individual wildebeest using *K*-means clustering. As such, the centroids of the segments were extracted and individual wildebeest were separated (Supplementary Fig. 1). The number of clusters in each segment was determined automatically by the ceiling division result of the number of pixels within the segment by the general wildebeest object size (namely, 9 pixels).

## Model evaluation

We evaluated the accuracy of the U-Net-based wildebeest detection model based on the alignment between the predicted wildebeest points and the ground reference points. A small local searching region was considered while matching the points to compensate for a slight shift, considering that the wildebeest segments were not always perfect $3 \times 3$ squares and the extracted centroids of the ground reference and predicted segment may not be perfectly aligned, but still represent the same animal. In this way, the extracted wildebeest centroids can still represent the correct detection of wildebeest even if they deviate by one pixel away from the ground reference points. The radius of the searching region was set to be 0.71 m, which is equivalent to the actual length of the diagonal line of one 0.5 m-resolution pixel. Predicted points that could be matched with one of the closest ground reference points within the searching region were counted as True Positive predictions. Predicted points that could not be matched with any ground reference points within the searching region were treated as False Positives, and all the remaining ground reference points that were not matched with any predicted points were treated as False Negatives.

To assess the overall performance of the model quantitatively, we utilized the following accuracy metrics: precision, recall and F1-score. Precision measures the accuracy of predicting wildebeest amongst all positive detections. It is calculated as the ratio between the number of True Positives and all detected positives. Recall measures how well the model performs at finding the actual true positives from all the ground reference points. It is the ratio between the number of detected True Positives and all existing ground reference positives. F1-score is the harmonic mean of precision and recall, which reflects the overall accuracy. The accuracy of each year was evaluated separately on the test dataset of each year, and the total accuracy obtained on all the test datasets was assessed as well. We repeated the model training and evaluation five times to obtain the uncertainty of the model accuracy.

In addition to the above, we adopted the precision-recall curve and area under the curve (AUC) to compare the performance of the sub-models with the U-Net-based ensemble model. By applying different thresholds to the probability map, we calculated multiple pairs of precision and recall. For the threshold of 0 or 1, we set the paired precision and recall rates as (0, 1) and (1, 0), respectively. These precision-recall pairs were then added to the plot, and AUC was calculated using the composite trapezoidal rule. The value of AUC is between 0 and 1. A larger AUC indicates better model performance.

To test the spatial and temporal transferability of the model, we ran two tests: (1) transferring the model to a temporally different dataset: we set aside the dataset in 2015 as an independent test dataset and trained the wildebeest detection model using only the data of the other five years (2009, 2010, 2013, 2018, 2020). The 2015 dataset is therefore an entirely new dataset obtained by a unique sensor with a different spatial resolution from others (38 cm of WV03 versus 42−50 cm of GE01 and WV02); (2) transferring the model to a spatially different dataset: we set aside the dataset in 2020 as an independent test dataset and trained the wildebeest detection model using only the data of the other five years (2009, 2010, 2013, 2015, 2018). The coverage of 2020 data is on the east side of the Masai Mara National Reserve and Serengeti National Park, which is outside the coverage of

the remaining datasets, and its spatial resolution is the coarsest (50 cm of WV02) among all the years. In each of the scenarios, the model was trained with datasets of five years and transferred to another new year with unseen features, such as new spectral characteristics of a different year, new image resolution and new landscapes. The model transferability in these two tests was evaluated directly using the test dataset of the independent year (2015 or 2020).

## Detecting and counting the wildebeest

After the U-Net-based ensemble model demonstrated a high accuracy using the test dataset, we applied the model to all the satellite imagery to detect all the wildebeest across the study area inside the Serengeti-Mara ecosystem. The images were cropped into patches to match the input size of the model, and the ensemble model outputs were converted using *K*-means clustering to obtain wildebeest point predictions. The detected wildebeest were then mapped across the study area. We counted the number of wildebeest points on each satellite image to obtain the population estimates. We repeated model training five times and calculated the count five times to obtain the associated modeling uncertainties (at a 95% confidence level) for each date.

To explore the spatial distribution patterns of the migrating wildebeest on different dates, we generated a point density map with a cell size of 100 m and a radius of 500 m (Fig. 8) for each date. The point density map visualizes the density of wildebeest points within the neighborhood of each pixel, showing the spatial and temporal variation in wildebeest distribution. We also calculated the wildebeest count per km$^2$ and summarized the frequency of the density as a histogram in Fig. 7.

## Reporting summary

Further information on research design is available in the Nature Portfolio Reporting Summary linked to this article.

## Data availability

The minimum set of segmentation mask samples that can be used to demonstrate the U-Net-based wildebeest detection framework generated in this study was deposited in the Github repository (https://doi.org/10.5281/zenodo.7810487). Samples of satellite images for model training and testing are available on a restricted basis due to data protection laws and access may be obtained by contacting the corresponding author upon reasonable request. The very-fine-resolution commercial satellite image data for wildebeest detection are protected under a NextView Imagery End User License Agreement and are not available as a result of data protection laws. The copyright remains with Maxar Technologies (formally DigitalGlobe), and redistribution is not possible. The detected wildebeest point data are available at: https://doi.org/10.5281/zenodo.7810487. Other data generated in this study to support the findings are provided in the Supplementary Information and Source Data File. Source data are provided with this paper.

## Code availability

The wildebeest detection framework based on U-Net is publicly available at Github repository[78] (https://github.com/zijing-w/Wildebeest-UNet); support and more information are available from Z.W. (zijingwu97@outlook.com).

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

## Acknowledgements

We thank Maxar Technologies (formerly DigitalGlobe) for providing very-fine-resolution commercial satellite images through the NextView Imagery End User License Agreement of the US National Geospatial-Intelligence Agency. We also thank the Army Departmental Requirements Office for acquiring the Maxar images for this work. We are grateful to Dr. Juan Lavista Ferres the Chief Scientist at the Microsoft AI for Good Research Lab for his support in our application for the Microsoft AI for Earth grant. Thanks also to Dr. Caleb Robinson and Dr. Anthony Ortiz for their assistance in transferring and storing satellite image data. We also wish to thank Ralph Mettinkhof for his support in using the Virtual Research Environment at the University of Twente. We are grateful to Dr. Olga Isupova and Dr. Xiaowen Dong for providing valuable comments on an early draft of this manuscript. Many thanks to Zeyu Xu for his support in testing YOLOv4 algorithm. L.F.H. acknowledges the support of the Ohrstrom Family Foundation.

This collaboration was supported, in part, by grant no. 00138000039 from the Microsoft's AI for Earth Program (https://www.microsoft.com/en-us/ai/ai-for-earth). Z.W.'s research was supported, in part, by funding from the Department of Natural Resources at the Faculty of Geo-Information Science and Earth Observation (ITC) of the University of Twente. This research was performed while I.D. held an NRC Research Associateship award at United States Army Research Laboratory. A.K.S.'s research was partly supported by funding from the European Research Council (ERC) under the European Union's Horizon

2020 research and innovation program (grant agreement no. 834709 BIOSPACE).

## Author contributions

T.W. and I.D. took the lead in organizing this collaboration. Z.W., T.W., and I.D. conceived the idea and designed the research. T.W. and A.K.S. supervised the project. Z.W. wrote the code and performed the computations and analysis with input from C.Z., X.G. and T.W. I.D., S.J.L., L.F.H., and J.A.S. acquired the satellite images. Z.W., T.W., C.Z., and X.G. prepared the training and test dataset. L.F.H., J.A.S., P.M.A., J.G.C.H., D.J.M., R.L., and S.N. interpreted the results. Z.W., I.D., L.F.H., and T.W. prepared the draft with substantial input to the science and manuscript from all authors and finalized the manuscript with the oversight from P.M.A.

## Competing interests

The authors declare no competing interests.
