## [Peer Review File · Nature Communications]

Deep learning enables satellite-based monitoring of large populations of terrestrial mammals across heterogeneous landscapesREVIEWER COMMENTS

Reviewer #1 (Remarks to the Author):

The authors employ a U-net for large migratory mammal surveys in high-resolution satellite imagery. The task of surveying wildlife populations more efficiently has been a major research focus for quite some time; recent developments using remote sensing (from drones to satellites) and machine learning and computer vision methodology have yielded great progress in this regard.

This study claims to be surveying wildlife at unprecedented scales. While this is not false per se and numerical results are promising, those are inferred from a very small portion of the data (less than 2% of the area); the remainder is either used for training (0.7%) or merely predicted on without quantitative analysis nor in-depth interpretation and discussion. The methodology itself is not groundbreaking enough to justify a lengthy discussion on it either. I thus recommend re-targeting some of the claims and shifting focus more towards ecological implications of the results (and the prospects of using satellite imagery and deep learning for surveys). Please see major and minor (line-specific) comments below.

MAJOR COMMENTS

- While results seem convincing, I am lacking an appropriate embedding via a discussion. The majority of the paper pre-Methods section is, as stated in the section title, "Results". However, since the methodology is not exactly revolutionary and these days well-established (deep learning-based species detection in remote sensing imagery has attracted quite some research already in the past), I would have been more interested in seeing the connection to, perhaps embedding in, ecological implications of the "large-scale" survey pursued here: do the trends in animal counts correspond to patterns observed in this migration event in other works? What can we read out of the significant differences in the number of animals detected across years, seasons, etc.?

- I am not sure about the decision on performing pixel-wise annotation + prediction, followed by individual disentangling by clustering. Of course, if an individual only covers a few pixels in area, performing object detection (via bounding box regression) seems not very sensible. However, using segmentation introduces a discrepancy between the human annotation task (it makes little sense to label individual pixels, hence the requirement of annotating points) and requires the extra step of having to employ k-means. This is particularly problematic, as the number of individuals (i.e., hyperparameter k) must be inferred or estimated by yet another heuristic, as it has to be set a priori (the chosen strategy is sensible, but assumes a standard animal size and resolution, which is not always the case).

- I am having troubles with the train/val/test splitting procedure. The Methods section (p.14, l.448ff) state that test sets were deliberately biased to include "sufficient wildebeest". In one sense, this is ok to assess model performance in one type of challenging situations (i.e., highly crowded scenes). But, as said, this is just one type; the other is performance in large amounts of empty (i.e., animal-free) scenes. Here, the risk is high for heterogeneous backgrounds to yield high numbers of false positives, and hence low precision – for such models to be applicable to large regions (as claimed in the introduction and discussion), they must be able to cope with this.

- Furthermore, the claim of this study having examined "unprecedented spatial scale(s)" "unlike previous attempts" (p.12, l.319f) is stretching it a lot, since the actual sizes of the training (0.7% of the whole area) and testing (1.7%) datasets (p.14, l.443ff) are minuscule. Sure, you eventually used the model to predict the whole area and produce density maps—whether or not these make any sense is not explained (related to comment above about discussion on ecological implications above). Unless I severely misunderstood the data setup I strongly recommend removing those sensational claims, or else expand the test set towards the full (remaining) area and redo the performance analysis accordingly.

Given the major workload induced by the latter, I recommend down-toning claims – this is perfectly fine, as you yourselves provide a (good) discussion that concentrating on known wildlife hotspot areas is good enough.

- About the ensembling strategy: in principle, this is a good idea as it has shown to increase robustness to a certain degree, even for deep learning models. It would be interesting to see how the ten-fold ensemble compares to a single model trained on all data together, though.

- I feel like a lot of references on past work are missing. Some of the methodologies used could

benefit from literature (e.g., Dropout); also, the number of works on aerial wildlife detection seems low.

MINOR COMMENTS

- p.1, l.40: drop "accuracy" and use "F1 score" exclusively. You might want to add precision and recall for a more fine-grained performance summary.
- p.2, l.53ff: every single one of these limitations also applies to image-based surveys (with or without machine learning assistance). The key advantages of imaging surveys over manned aerial or foot surveys are (i.) precise spatial localisation ability, (ii.) increased robustness due to possibility of having multiple observers annotate images, (iii.) lower cost and risks for humans and wildlife (UAV and satellite-based); no variation in height and speed (satellite-based). As opposed to other survey methods, satellites introduce another potential issue, though: occlusion due to clouds.
In short, I would re-target the first introduction paragraph towards limitations your proposition actually promises to overcome, or else elaborate on the issues you raised and how your method remedies them.
- p.2, l.56: (observation, no action required) I must have read at least six different variants on acronyms for drone and their meaning. "Unoccupied" aerial vehicles is yet another one I haven't seen before.
- p.2, l.59: most recent drones (and cameras) can acquire subdecimeter resolution imagery (sometimes down to a few cm) and fly at >200m above ground. I have yet to see a paper on it, but personal observation has shown these systems not to cause any disturbance to wildlife whatsoever anymore.
- p.2, l.65: again nitpicking, but size of animals isn't always a reason to go for proxies. An Orangutan nest is about the same size as the individuals, yet detecting the latter beneath the canopy is near-impossible.
- p.2, l.67: manual detection, lack of transferability in models *and* the high cost of high-resolution satellite imagery have prevented large-scale studies—and they still do.
- p.2, l.80ff: this statement is incomplete/confusing until the reader realises you are exclusively talking about wildlife detection in satellite imagery. In drone data, for example (which also counts as "remotely sensed images", cf. line 79), very small animals such as seabirds have also been successfully detected, and even so without high contrast and/or homogeneous environments. Perhaps cite those as well, or emphasise you are talking about satellite-based species mapping.
- p.2, l.90: perhaps you want to add the original reference for U-net here (Ronneberger et al., 2015)? Also, the choice of a semantic segmentation model for object detection seems at first kind of counterintuitive. It does make sense if the resolution of satellite imagery is coarse enough for e.g. one pixel to encompass a single individual. That would have to be clarified, though.
- p.3, l.103ff: I was interested in this claim and looked it up in the cited refs. I believe a slight clarification on the ways in which climate change affects this seasonal migration event is something the general audience of this journal would want to read about, too. Maybe add one embedded sentence about that?
- p.3, l.123ff: I understand why you put this sentence, but it takes up a lot of space for a detail. I would move it somewhere else if possible.
- p.4, Fig.1: I believe this figure could benefit from some re-styling. For example, the satellite patch is very small and details thus hard to see (despite the enlargement in the vectorisation part); the U-net flowchart seems a bit odd (intermittent conv and pool layers aren't shown); "vectorisation by K-means clustering" makes no sense and contradicts the caption and text; the colour scheme is a bit wild. Also, minor detail: I would reformulate the caption into a passive style (i.e., "label the wildebeest (...)") → "wildebeest are labelled (...)").
- p.4, l.146ff: very good. But this automatically implies that your model will only work for "high-contrast animals and homogeneous landscapes", precisely one of the limitations of existing approaches presented in the motivation part of the introduction.
- p.4, l.149: please specify that humans annotated points here, not individual pixels (otherwise the number of annotated wildebeest would be hard to state).
- p.4, l.152: what does "manual temporal image differencing" mean? Please elaborate.
- p.4, l.153: "we build a test dataset on each image" – that makes no sense. TODO
- p.5, l.162ff: maybe specify that precision and recall were calculated on a per-individual and not per-pixel basis.

- p.5, l.164: "2700 test image *patches*" – that sounds like a very low number, especially given the size of the study area and number of time steps. Furthermore, what is a "patch" in your setting?
- p.5, l.173: "Fig. 1 shows the variation in the ensemble model arising from splitting the training dataset" – no it doesn't, there's virtually no differences visible between the three (random) splits shown in that figure.
- p.5, l.177: "summarized" → "averaged"
- p.7, Fig.3: word has to be taken for granted that these red circles really are on wildlife individuals. This is very hard to see. Perhaps you can add (in other colours) false positives and false negatives? I find it hard to believe that the model made so little mistakes (cf. high precision and recall), given the minuscule individual size and background patterns that could be mistaken for animals at that scale.
Also, these red circles are generally visible, but in the bottom right figures they vanish when simulated with red-green-blindness vision. Maybe choose a more contrastive colour?
- p.8, l.247ff: "Although the datasets were for different years" – I am wondering what this statement would tell to the reader. Detecting half a million individuals is impressive, but as you state this is a cumulative and thus highly inflated number. Looking at Table 1, the variance in detection count is enormous. Sure, much of this fluctuation can be attributed to different months of acquisition and climatic and other effects having occurred across the different years. It would have been great to see results across all years in imagery taken at the same time of the year each time.
- p.8, l.256ff: follow-up: this is interesting, but one crucial aspect is missing: the discussion. Do these spatiotemporally characterised wildlife densities correspond to independent observations made about migration dynamics in the area?
- p.11, Fig. 5: can you make these hotspot clusters semi-transparent? Why did you use a different background dataset than the images you already have? Also, I assume you used kernel density estimation to create these hotspots? You might want to at least put the term in the caption.
- p.12, l.323ff: "tremendous" spectral diversity is a bit of an overstatement, no? Also, this by itself is no warrant that generalisation to both new species and landscapes be feasible – if there's a domain shift it has to be accounted for; more variance in descriptors don't automatically make up for it. Furthermore, you explain the methods that you only keep the four "usual" bands (RGB-NIR), nullifying this advantage.
- p.12, l.331f: here's the Achilles' heel of the proposed workflow: the method is (still) extremely expensive due to the requirement of high-resolution satellite imagery.
- p.12, l.332ff: since you mention statistical methods on top of manual surveys (rightfully, as they are still the de facto standard for many conservation areas), it would have been interesting to include count estimations from them via spatially and temporally overlapping surveys. Do you have such data, at least for parts of the years/regions you included?
- p.13, l.373: "this is an advancement" – again, playing devil's advocate I dare say this is a bit of an overstatement. Sure, object detectors struggle with such small animal sizes as mentioned, but the proposed method introduces other challenges (separation of two individuals close together isn't always guaranteed; number of clusters needs to be known a priori, etc.).
- p.13, l.378ff: given that it is already non-trivial to disentangle some species in very high-resolution UAV imagery, expecting this to be possible at a fine grain in satellite data is not realistic.
- p.14, l.428: does this mean that each animal was annotated with a 3x3 pixel square? I don't see the point in that (no pun intended).
- p.14, l.429ff: doing this for inter-observer reliability is great for sure. I would be interested in both qualitative and quantitative measures on how high the agreement was: how big is the percentage of the intersection of the annotations the four experts agreed on, in comparison to all annotations made? How hard was it to spot the animals (the description of animal appearances are all good, but having experts' opinions on the feasibility of annotation would be quite helpful here, also given the subjectively high difficulty of the task shown in Extended Data Figure 2).
- p.14, l.439: was there a reason for this particular patch size?
- p.14, l.444ff: how much in percentage was the test dataset, then? This sounds like it is less than 0.15%, even though you later state (l.467) that it is actually 1.7%, again confirming a high spatial bias. Also, "stratified random sampling method" implies this is not the test, but the validation set – please specify.
- p.15, l.489: maybe write "low-resolution but high-dimensional" to explain this rationale to non-

expert readers.

- p.15, l.495ff: this is not clear enough: did you use a softmax activation and cross-entropy loss for classification (which is the norm) or did you apply a sigmoid activation and use binary cross-entropy losses instead?
- p.15, l.509: would it be possible to add an equation of the Tversky loss? That also makes it easier to understand the alpha and beta parameters.
- p.15f, l.517f: so the 2009 dataset serves as validation here? Why this particular year? Your entire train/val/test split setup seems increasingly confusing, to be honest.
- p.16, l.534f: how did you do this normalisation? Softmax? Sigmoid? Something else?
- p.16, l.537: it seems strange that the threshold chosen is precisely 0.5. In such an imbalanced setting I would have expected it to be different. Did you tune it on a validation set?
- p.18, l.632ff: is there a reason why you did not publish your code on a dedicated site for open source projects like GitHub?
- Extended Data Fig. 3: this is a more interesting figure than Figure 2 in the main text, for reasons explained above (see minor comment): it also shows false positives and negatives. The difficulty of the scenes (esp. bottom row) is rather low, though.
- Extended Data Figure 6: this colour scale is... useless, really; there's no way one can identify performance differences this way.
- Extended Data Fig. 7: to be frank, tuning an evaluation criterion for maximum performance does not sound legit. It would have been better to motivate this choice by prior assumptions, e.g. on the expected average length of individuals, rather than to beautify model results by choosing the radius yielding highest performance readings.

Reviewer #2 (Remarks to the Author):

This is an important paper in the growing field of counting large animals from space with satellite remote sensing. I encourage its publication.

The U-Net-based ensemble learning model approach outlined here represents (to my knowledge) an important next step in the remote sensing of animal abundance. It will likely prove useful in tracking the migrations of other large ungulates (pronghorn, caribou, bison, saiga, Tibetan gazelle, etc.). Maintaining these increasingly threatened migrations around the world is critical for continuing the ecological functioning of some of the last wild places on the planet. Getting reliable estimates of the abundance of keystone species on the move from space is a critical step for developing a more predictive ecology that will improve conservation and land management. Furthermore, it will do so in a manner that should be cheaper and less disruptive to the animals counted than current measures.

I found the paper to be quite thorough in terms of its description of the entire pipeline from satellite observations through model training and testing, validation and performance to postprocessing and detection. The discussion of model transferability both temporally and spatially was welcome. This is a fairly large study in terms of space (2747 sq. km.), time (images from 6 years across a total time frame of 12 years), total animals counted (~480,362), and the number of space sensors used (three satellites operating at different ground spatial resolutions). The figures are quite helpful (in addition to those in the paper, I found Extended Data Figures 1 and 2 to be especially useful). Their methods appear to be novel, sound, and sufficiently detailed.

One issue: Unless I'm missing something, I believe all of the training, ground reference, and validation data come from the same satellite datasets used in the analyses themselves. Given the spatial scales covered, I understand this. Nevertheless, I suspect that the community of ecologists and field biologists who have spent decades counting large ungulates on the move from ground counts and aerial platforms would like to see subsets of ground and/or aerial imagery used in the validation of the wildebeest counts published in this paper. I don't believe this constitutes a serious flaw in the analyses presented but do think future applications might consider how to incorporate samples from more traditional approaches for intercomparison with the results of the U-Net-based model. I realize scaling challenges will make this difficult but believe the links could still be made through incorporation of subset counts from more and less dense aggregations of wildebeest using traditional methods.

A minor point: at a couple of places in the article wildebeest are referred to as “small-bodied” animals. I think animals of 1.5m to 2.5m don’t really qualify as small bodied—even among mammals.

There appears to be a minor typo in line 72 at the top of page 10 (in the caption for Figure 4) which refers to wildebeest detected in August 2019. I believe this should be August 2018.

I like this work and wish NASA had funded it.

Reviewer #3 (Remarks to the Author):

Review for: “Satellite-based monitoring of the world’s largest terrestrial mammal migration using deep learning”

This manuscript presents a machine learning approach to count individual ungulates in commercial satellite imagery. The approach seems to work well across imagery from different satellites, with slightly different spatial resolutions, across distinctive habitats – which is impressive. Overall, this work presents an important first step in developing approaches to leverage commercial satellite imagery for monitoring wildlife populations. I have expertise in animal movement and migration ecology, but I am not an expert in machine learning. Therefore, I have prepared my review as a potential user of this approach and cannot speak to the technical soundness of the approach presented in the manuscript. I was enthusiastic to read this manuscript, because this technique would be useful and relevant in my research. However, I do have some concerns about the manuscript and the utility of this approach for monitoring wildlife and understanding animal behavior.

Main concerns/points to consider:

Essentially, this is a proof of concept that commercial satellite imagery can be used to count herds of ungulates. The main outcomes of this approach are 1) counts derived from imagery on 6 days during different parts of the migration cycle, 2) heatmaps of ungulate density and 3) the analytical pipeline. No further analyses are done and no further ecological insights are drawn from this data. So, as I see it, the key contribution of this manuscript is the development of a new method to count wildlife using submeter-resolution satellite imagery. Yet, although the main contribution is methodological, the format of Nature Communications presents all methodology details at the very end of the manuscript and within the supplements. This misalignment of the core contribution of the paper and the formatting requirements of Nature Communication seems odd and potentially problematic. For this reason, Nature Communications might not be an ideal fit for this manuscript. However, I will leave this up to the editor to determine. I think there is wide enough interest in this topic, that it could warrant publication in a high-impact journal. I am just not sure that the methods last format of Nature Communications is logical for this contribution.

Next, how were the dates of the imagery chosen? The selection of August 2009, September 2010, August 2013, July 2015, August 2018, and October 2020 seems somewhat random. Is this because those were what images were readily available within your area of interest or was there some other reason for picking these dates? More information on temporal resolution and availability of data is needed to assess the utility of this method for broad application in ecology and wildlife monitoring. It would be useful to know if the temporal resolution of image availability is a limiting factor, because this would affect the types of questions researchers can address and how widespread and useful this new method would be.

How ready to use is the software/code in Dryad? A main contribution of the work is the analytical pipeline, so having usable code would be a huge contribution. I couldn’t tell, because the link to the dryad repository isn’t active/accessible yet.

Transferability How transferable is this pipeline to other datasets? In lines 323-325 of the discussion, it is mentioned that the scope and size of this project makes it so that this model could

be easily generalized to new systems and species. I think anyone interested in generalizing this model to their system would be curious to know how involved this process would be. For example, to use this approach on caribou in the artic, would you need to annotate new training data? How much? How would you recommend iterating and tweaking the model? This type of information would be useful to potential end-users looking to adopt your approach.

Validation/labeling data – how consistent were the 4 annotators that labeled the training dataset? I am curious to know if the level of human error is similar to that of the deep learning algorithm? Since there is no independent way to validate the counts (besides using human-annotated data from the same commercial images), I think more exploration could be warranted.

How useful would this approach be for monitoring a migratory population if the migration pattern is unknown? In many places, where animals move is not well documented. And as we place more tracking devices on animals, scientists are frequently discovering more surprising new migration routes. How do you see this approach fitting in more broadly with the use of other tracking technologies?

Minor points:

I found it slightly confusing that in the abstract you say that the method counts large herds of migratory ungulates (wildebeest and zebras). Then, all of the figures and results are labeled specifically as wildebeest detections. Finally, when we move back into the discussion section, there is an entire paragraph that acknowledges that it is not possible to distinguish between wildebeest and other similarly sized ungulates. So I wonder, if the way the results are labeled and presented should be modified to reflect this uncertainty.

Lines 109-113 of the introduction – What exactly do you mean by “this technology”. Is it the automated approach to counting wildlife using commercial satellite images? Or the ability to frequently and accurately assess the status of migratory ungulate populations? I think there are still too many dots to connect between an automated approach to counting animals in satellite imagery and understanding emergent properties of animal groups. For example, I think you would need more frequent data collection, and the ability to differentiate between individuals across time periods to truly get at any of these emergent phenomena. I suggest either deleting this sentence or pairing it back to be more relevant to the ecological questions your analytical pipeline can help to tackle now.

Figure 3 – is it possible to include the zoomed-in squares in Fig.3 A-E without the red circles? It is hard to see what the algorithm is picking up on with the image modified like this. I think having both the annotated and unannotated images side by side would be a helpful visualization.

Reviewer #1 (Remarks to the Author):

The authors employ a U-net for large migratory mammal surveys in high-resolution satellite imagery. The task of surveying wildlife populations more efficiently has been a major research focus for quite some time; recent developments using remote sensing (from drones to satellites) and machine learning and computer vision methodology have yielded great progress in this regard.

This study claims to be surveying wildlife at unprecedented scales. While this is not false per se and numerical results are promising, those are inferred from a very small portion of the data (less than 2% of the area); the remainder is either used for training (0.7%) or merely predicted on without quantitative analysis nor in-depth interpretation and discussion. The methodology itself is not groundbreaking enough to justify a lengthy discussion on it either. I thus recommend re-targeting some of the claims and shifting focus more towards ecological implications of the results (and the prospects of using satellite imagery and deep learning for surveys). Please see major and minor (line-specific) comments below.

Response: We appreciate the insightful and helpful comments from the reviewer which allowed us to further improve our manuscript. We present our detailed responses to your major and minor comments below.

MAJOR COMMENTS

- While results seem convincing, I am lacking an appropriate embedding via a discussion. The majority of the paper pre-Methods section is, as stated in the section title, "Results". However, since the methodology is not exactly revolutionary and these days well-established (deep learning-based species detection in remote sensing imagery has attracted quite some research already in the past), I would have been more interested in seeing the connection to, perhaps embedding in, ecological implications of the "large-scale" survey pursued here: do the trends in animal counts correspond to patterns observed in this migration event in other works? What can we read out of the significant differences in the number of animals detected across years, seasons, etc.?

Response: We agree that the U-Net architecture is a well-established deep learning algorithm. However, directly applying U-Net, or alternative popular object detection deep learning algorithms (e.g., YOLO) to automate the accurate location and counting of hundreds of thousands of individual small animals across a large, highly heterogeneous landscape is still very challenging. That is why we developed a novel automated machine learning pipeline (framework) that involves an ensemble model based on a U-Net deep learning network followed by a post-processing module. Our study shows that the pipeline is robust and transferable, and achieved highly accurate results with an overall F1-score of 84.75%. To the best of our knowledge, this is the first demonstration of deep learning techniques to monitor wildebeest-sized animals across an area of this size by counting hundreds of thousands of individuals in highly heterogeneous background context. Our approach holds promise for scaling to produce the first ever total count of wildebeest or other ungulates in open landscapes e.g., white-eared kob in South Sudan. In addition to facilitating total counts for multiple species, the ability to

observe expansive herds of migratory ungulates from satellite sensors presents an exciting opportunity for the study of the ecology of animal aggregations from an entirely novel perspective. In this regard, we believe that our study is novel in scale, scope, and technique. Nevertheless, we agree with the suggestion and have re-targeted some of the claims and shifted focus more towards ecological implications of the results. For further details please refer to our revised manuscript (see page 7, lines 300-355).

- I am not sure about the decision on performing pixel-wise annotation + prediction, followed by individual disentangling by clustering. Of course, if an individual only covers a few pixels in area, performing object detection (via bounding box regression) seems not very sensible. However, using segmentation introduces a discrepancy between the human annotation task (it makes little sense to label individual pixels, hence the requirement of annotating points) and requires the extra step of having to employ k-means. This is particularly problematic, as the number of individuals (i.e., hyperparameter k) must be inferred or estimated by yet another heuristic, as it has to be set a priori (the chosen strategy is sensible, but assumes a standard animal size and resolution, which is not always the case).

Response: We agree that if an individual only covers a few pixels in an area, performing object detection (via bounding box regression) does not seem sensible indeed. This is the case for the wildebeest-sized objects which cover only a few pixels (≤ 9 pixels) in the area. We, therefore, decided to perform pixel-wise annotation and prediction, followed by individual disentangling by clustering. Nevertheless, we have conducted an extra experiment using a deep learning-based object detection algorithm (YOLOv4) within this study, and trained the model with the same dataset. The method resulted in an overall F1-score of 72.13% (with a precision of 76.95%, recall of 67.88%). Our method clearly performs much better than the object detection algorithm even with varying image resolution.

Indeed, our deep learning pipeline includes an “additional” step of segmenting the detected objects, but this step of determining parameter k in k -means is a fully automatic process in this pipeline once the user determines the general size of each animal (in this study it is a 3 by 3 pixel segment). k is then calculated by: $k = \text{ceil}(\text{segment size} / 9)$. The step of setting the animal size is not arbitrary because our pipeline is designed specifically for relatively small-bodied animal detection (compared with very large animals such as elephants and whales) on sub-meter satellite imagery, and the target animals in general have a similar size in the imagery. The wildebeest appearing in the satellite images are not always exactly 3 by 3 pixels, and they usually are smaller (or, in some rare cases, bigger) depending on their moving direction and their body shadow as well as the resolution of the satellite images (38-50 cm). Empirically, in this study, the size 3 by 3 is effective even with varying animal sizes.

- I am having troubles with the train/val/test splitting procedure. The Methods section (p.14, l.448ff) state that test sets were deliberately biased to include "sufficient wildebeest". In one sense, this is ok to assess model performance in one type of challenging situations (i.e., highly crowded scenes). But, as said, this is just one type; the other is performance in large amounts of empty (i.e., animal-free) scenes. Here, the risk is high for heterogeneous backgrounds to yield

high numbers of false positives, and hence low precision – for such models to be applicable to large regions (as claimed in the introduction and discussion), they must be able to cope with this.

Response: Agreed. The performance of the model on wildebeest-abundant scenes and wildebeest-free scenes are both important, which is the reason why we adopted stratified proportionate random sampling based on the wildebeest density to ensure the test set included scenes with no wildebeest as well as those highly crowded with wildebeest. In fact, the test set contains sample grids with four different levels of wildebeest density (low density, medium density, high density and very high density) (see page 10, lines 448-450, lines 455-457). We provided one example in Supplementary Figure 9 showing varied wildebeest density distributions across different areas in the image of year 2009, and the majority of the image contains a low density of animals. To make it clearer, we have also added Supplementary Figure 2 showing the distribution of the number of animals on the image patch in the test and training dataset. The majority of the test images have a low number of animals. In fact, there are 2335 test image patches with no wildebeest in the image.

- Furthermore, the claim of this study having examined "unprecedented spatial scale(s)" "unlike previous attempts" (p.12, 1.319f) is stretching it a lot, since the actual sizes of the training (0.7% of the whole area) and test (1.7%) datasets (p.14, 1.443ff) are minuscule. Sure, you eventually used the model to predict the whole area and produce density maps—whether or not these make any sense is not explained (related to comment above about discussion on ecological implications above). Unless I severely misunderstood the data setup I strongly recommend removing those sensational claims, or else expand the test set towards the full (remaining) area and redo the performance analysis accordingly.

Given the major workload induced by the latter, I recommend downtoning claims – this is perfectly fine, as you yourselves provide a (good) discussion that concentrating on known wildlife hotspot areas is good enough.

Response: We agree that the tone of some of our claims were unnecessarily sensationalist and might have created some distractions, unfortunately. We do believe that the volume and the animal sample size of the training and test data included in these exercises is above the average in the literature (e.g. Brandt et al.: 100 sample plots in test set; Gonçalves et al.: 10,766 training points and 1,168 animals in test set), but we take the point and admit that these kinds of claims will only serve to distract the reader.

Following your suggestion, we conducted a full review of the manuscript and have toned down these claims and language. As an example of these modifications, the sentence was changed from:

“Unlike previous attempts to automate this process, we deployed our model at an unprecedented spatial scale (2,747 km²) and validated it on a dataset that varied in space, time, and resolution.”

to

“To create outputs that would have real-world utility to researchers and managers, we deployed our model at an especially large spatial scale (2,747 km²) and validated it on a dataset that varied in space, time, and resolution.” (see page 6, lines 289-291).

References:

Brandt, M., Tucker, C.J., Kariryaa, A., Rasmussen, K., Abel, C., Small, J.L., Chave, J., Rasmussen, L.V., Hiernaux, P., Diouf, A.A., Kergoat, L., Mertz, O., Igel, C., Gieseke, F., Schöning, J., Li, S., Melocik, K.A., Meyer, J., Sinno, S.S., Romero, E., Glennie, E., Montagu, A., Dendoncker, M., & Fensholt, R. (2020). An unexpectedly large count of trees in the West African Sahara and Sahel. *Nature*, 1-5.

Gonçalves, B. C., Spitzbart, B., & Lynch, H. J. (2020). SealNet: A fully-automated pack-ice seal detection pipeline for sub-meter satellite imagery. *Remote Sensing of Environment*, 239, 111617.

- About the ensembling strategy: in principle, this is a good idea as it has shown to increase robustness to a certain degree, even for deep learning models. It would be interesting to see how the ten-fold ensemble compares to a single model trained on all data together, though.

Response: We also ran the model without using the ensemble approach and compared the performance against the ensemble approach. The results are presented in the texts: *“the F1-score of 10 base models on average is 78.22% ($\pm 0.86\%$), also lower than the F1-score of ensemble model (84.75%).”* (see page 5, lines 218-219), as well as in Figure 5B and Supplementary Table 4. We would like to clarify that our validation data were taken from the training dataset (10% of it). Therefore, the results of the individual base models actually reflect the model performance already without using ensemble.

- I feel like a lot of references on past work are missing. Some of the methodologies used could benefit from literature (e.g., Dropout); also, the number of works on aerial wildlife detection seems low.

Response: Thanks for this suggestion. We have added more references on ensemble learning (see page 11, line 502), dropout (see page 11, line 529), and previous work on aerial wildlife detection (see page 2, lines 67-69).

MINOR COMMENTS

- p.1, l.40: drop "accuracy" and use "F1 score" exclusively. You might want to add precision and recall for a more fine-grained performance summary.

Response: Agreed. We have changed “accuracy” to “F1-score” and added precision and recall: *“with an overall F1-score of 84.75% (Precision: 87.85%, Recall: 81.86%).”* (see page 1, lines 43-44).

- p.2, l.53ff: every single one of these limitations also applies to image-based surveys (with or without machine learning assistance). The key advantages of imaging surveys over manned aerial or foot surveys are (i.) precise spatial localisation ability, (ii.) increased robustness due to

possibility of having multiple observers annotate images, (iii.) lower cost and risks for humans and wildlife (UAV and satellite-based); no variation in height and speed (satellite-based). As opposed to other survey methods, satellites introduce another potential issue, though: occlusion due to clouds.

In short, I would re-target the first introduction paragraph towards limitations your proposition actually promises to overcome, or else elaborate on the issues you raised and how your method remedies them.

Response: We agree with the summary of the key advantages of imaging-based wildlife surveys over manned or foot surveys. We have revised these sentences to: “*However, crewed surveys introduce risks to human and wildlife and in many cases can only provide animal counts with coarse location precision. Moreover, all crewed aerial survey techniques are subject to biases arising from detection probability, observer experience and double counting*^{8,12}.” (see page 2, lines 64-67). Regarding the limitation of satellite-based survey, we also agree that satellites introduce the potential issue of cloud cover. However, weather condition is an issue for the crewed/uncrewed aerial surveys as well, such as wind and rain that can affect the feasibility of the flight operations and fog that can influence the visibility.

We agree to re-target the first introduction paragraph. Therefore, we revised and added a new first introduction paragraph in our manuscript and have oriented it towards the need for advanced monitoring techniques that can provide managers with information at a rate that can keep pace with local environmental changes (see page 2, lines 52-59).

- p.2, l.56: (observation, no action required) I must have read at least six different variants on acronyms for drone and their meaning. "Unoccupied" aerial vehicles is yet another one I haven't seen before.

Response: We have changed "unoccupied" to "uncrewed" to keep it consistent throughout the manuscript.

- p.2, l.59: most recent drones (and cameras) can acquire sub decimeter resolution imagery (sometimes down to a few cm) and fly at >200m above ground. I have yet to see a paper on it, but personal observation has shown these systems not to cause any disturbance to wildlife whatsoever anymore.

Response: Agreed. Previous study of our co-author Duporge and Wang (<https://doi.org/10.1111/2041-210X.13691>) have investigated the disturbance caused by drones to wildlife at different flight altitudes. However, the flight restriction still exists in some protected areas. We have revised this sentence as “*Moreover, UAVs can disturb wildlife when flown at a low altitude*²⁰⁻²², which has led to flight restrictions in some protected areas²³.” (see page 2, lines 71-72).

- p.2, l.65: again nitpicking, but size of animals isn't always a reason to go for proxies. An Orangutan nest is about the same size as the individuals, yet detecting the latter beneath the canopy is near-impossible.

Response: Agreed. The challenge of detecting animals at the individual level is not always the size. For example, in some cases the animals are occluded by vegetation or deep shadow. We have revised the texts to clarify this: “*Many of the first applications of this technology focused on visualizing and analyzing easier-to-view environmental markers that, in certain contexts, provide insights to estimate population size (e.g., guano stains²⁴, nests²⁵, mounds and burrows²⁶). It took less than a few years, however, for the technology to accommodate manual counts at the scale of individual animals for species in unobscured contexts (e.g., polar bears²⁷, albatrosses²⁸, and Weddell seals^{29,30}).*” (see page 2, lines 76-81).

- p.2, 1.67: manual detection, lack of transferability in models *and* the high cost of high-resolution satellite imagery have prevented large-scale studies—and they still do.

Response: Agreed. We have modified the texts: “*However, reliance on labor-intensive manual detection has restricted uptake by the conservation community, highlighting the need for automated techniques for processing fine-resolution satellite images.*” (see page 2, lines 81-83).

- p.2, 1.80ff : this statement is incomplete/confusing until the reader realises you are exclusively talking about wildlife detection in satellite imagery. In drone data, for example (which also counts as "remotely sensed images", cf. line 79), very small animals such as seabirds have also been successfully detected, and even so without high contrast and/or homogeneous environments. Perhaps cite those as well, or emphasise you are talking about satellite-based species mapping.

Response: Agreed. We have revised it and changed “*remotely sensed images*” to “*satellite imagery*” (see page 2, line 92).

- p.2, 1.90: perhaps you want to add the original reference for U-net here (Ronneberger et al., 2015)? Also, the choice of a semantic segmentation model for object detection seems at first kind of counter intuitive. It does make sense if the resolution of satellite imagery is coarse enough for e.g. one pixel to encompass a single individual. That would have to be clarified, though.

Response: Thanks for the suggestion. We have added the motivation of using semantic segmentation model and included the original reference of Ronneberger et al. for U-Net: “*At the same time, new collaborations between ecologists and computer scientists have provided several key advancements in automated animal detection from satellite imagery, including detection of the world’s largest marine and terrestrial vertebrates, such as whales³⁷ and elephants³⁸, using object detection algorithms (Faster-RCNN). However, current object detectors suffer from the small size of the objects in imagery^{39–41}. The feasibility of successfully using object detection methods is dependent on the body size of the animal: mature whales have a body length of more than 20 meters⁴², and the African elephants are approximately 3 to 4 meters in length⁴³, both of which have more than 8 pixels along the body length axis in satellite imagery.*”

A few studies have conducted automated surveys for smaller species using satellite imagery, such as for seals⁴⁴ and albatrosses⁴⁵ using pixel-based semantic segmentation algorithms like U-Net. Image segmentation deep learning architectures such as U-Net⁴⁶ predict the class probability for

every pixel, showing the potential to detect animals with a smaller size in satellite imagery.” (see page 2-3, lines 90-104).

- p.3, l.103ff: I was interested in this claim and looked it up in the cited refs. I believe a slight clarification on the ways in which climate change affects this seasonal migration event is something the general audience of this journal would want to read about, too. Maybe add one embedded sentence about that?

Response: We have added some texts and one more reference to clarify the influence of climate change on the migration: *“with the migration subject to seasonality of rainfall and habitat preference, this iconic system is facing unprecedented threats from rapid climate and environmental change⁵⁴⁻⁵⁷.”* (see page 3, line 127).

- p.3, l.123ff: I understand why you put this sentence, but it takes up a lot of space for a detail. I would move it somewhere else if possible.

Response: Agreed. On the other hand, we also need to make it clear before presenting the results so the reader will not find it confusing (see Reviewer #3’s comment). Thus, we have moved it to the end of this paragraph (see page 4, lines 169-172).

- p.4, Fig.1: I believe this figure could benefit from some re-styling. For example, the satellite patch is very small and details thus hard to see (despite the enlargement in the vectorisation part); the U-net flowchart seems a bit odd (intermittent conv and pool layers aren't shown); "vectorisation by K-means clustering" makes no sense and contradicts the caption and text; the colour scheme is a bit wild. Also, minor detail: I would reformulate the caption into a passive style (i.e., "label the wildebeest (...)” → "wildebeest are labelled (...)”).

Response: Thank you very much for the advice. We have updated the satellite image examples in this figure by enlarging the images for better visualization in Figure 1. We have also updated the chart of U-Net architecture by adding more details of the network, including the convolution, max pooling and up-sampling layers. We have also removed the description of “vectorisation by K-means clustering” and revised the corresponding texts in the caption and the main article. Regarding the passive style in the caption, we have revised the caption following this suggestion: *“Figure 1. **Model framework.** The wildebeest detection pipeline consists of three main blocks: 1) The wildebeest are labeled in the satellite imagery and the masks are generated; 2) The satellite images and the masks are fed into the U-Net-based ensemble model for model training/validation and to produce the wildebeest probability maps; 3) The probability maps produced by the 10 base models are averaged to obtain the final predictions and the wildebeest individuals are detected using K-Means clustering.”* (see page 21, lines 835-840).

- p.4, l.146ff : very good. But this automatically implies that your model will only work for "high-contrast animals and homogeneous landscapes", precisely one of the limitations of existing approaches presented in the motivation part of the introduction.

Response: One of the limitations of existing approaches (as described in our paper) is indeed that they only work for high-contrast species and homogeneous landscapes, for example, identifying seals from sea ice (where the dark color of pack-ice seals contrasts with the white sea

ice) and identifying albatrosses from nesting habitat (the white color of albatrosses contrasts with their green nesting habitat). In our study, we attempt to locate the grayish brown wildebeest that have a relatively low contrast to their complex habitat (e.g., mixed forest and savanna ecosystems). On the other hand, our model can accurately identify wildebeest across different types of landscapes with very high variation. We believe this is a wording issue because high or low contrast and homogeneity or heterogeneity are relative terms. To avoid confusion, we have modified our original sentence as “*Each individual wildebeest in the satellite imagery was represented by approximately 3-to-4 pixels in length and 1-to-3 pixels in width, with 1 or 2 relatively darker pixels in the center, including the shadow of the body.*” (see page 4, line 179).

- p.4, l.149: please specify that humans annotated points here, not individual pixels (otherwise the number of annotated wildebeest would be hard to state).

Response: Many thanks for the advice. We have revised the text to “*wildebeest points*” (see page 4, line 182).

- p.4, l.152: what does "manual temporal image differencing" mean? Please elaborate.

Response: We have revised this part and added a further clarification to the text: “*During the labelling process, we used a set of reference satellite images acquired on different dates, but with the same background landscapes for cross-referencing to ensure the labels were moving animals and were not similar-looking static objects (e.g., termite mounds, small bush).*” (see page 4, lines 184-187).

- p.4, l.153: "we build a test dataset on each image" – that makes no sense. TODO

Response: The statement “build a test dataset on each image” is indeed confusing. We have revised the sentence to “*To evaluate model performance, we used a stratified random sampling method to select test sample plots across the images in each year to ensure their representativeness and independence from the training dataset.*” (see page 5, lines 192-194). As the image scenes have different spatial and temporal coverages, we selected the test samples separately for each year: for instance, we selected 100 sample plots on the image of year 2009, and 200 sample plots on the images of year 2013 (see page 10, lines 457-467).

- p.5, l.162ff: maybe specify that precision and recall were calculated on a per-individual and not per-pixel basis.

Response: We have added one sentence to clarify this: “*The accuracy (precision, recall, F1-score) was evaluated on a per-individual basis as demonstrated in Fig. 4.*” (see page 5, lines 201-202).

- p.5, l.164: "2700 test image *patches*" – that sounds like a very low number, especially given the size of the study area and number of time steps. Furthermore, what is a "patch" in your setting?

Response: These 2700 test image patches cover approximately 1.7% of the whole dataset in terms of the area covered. And the number of animals in the test dataset is about 2.4% of all the animals in the whole dataset. We do think this sample size is arguably above average compared

with other studies (e.g. Brandt et al.: 100 sample plots in test set from a whole dataset of 1.3 million km²). We aim to evaluate the performance of the model using a subset of the data, and the feature space of this subset (the test set) in general should be rich, and represent the whole image set. This is why we adopted stratified random sampling - to ensure that the test set is representative of the whole dataset in terms of the complexity (varied levels of animal density) (see page 10, lines 455-457). As a counter-example, if we randomly took half of the whole image dataset as the test set, but this half happened to cover no wildebeest (it is obvious from Figure 7 and 8 that most of the area covers very small numbers of wildebeest), the accuracy would only reflect how good the model is to avoid false positives, while showing little about the ability to accurately detect animals. Moreover, the population size does not drive the statistics – for example, in the classical case of estimating the mean by sampling from a large, but finite population, the sample size required is independent of population size, and rather dependent on the standard error desired and the standard deviation of the data (assuming independent and identically distributed data). Here, the goal is not to estimate the mean, of course, but the same principle holds true, that the result is not dependent on the population. What matters is that the algorithm is able to learn sufficiently across the (feature) space of interest, and our results suggest that this is indeed the case, and our spatial sampling strategy was designed specifically to achieve this. In any case, we believe that our sampling ratio is large compared with other studies.

Regarding the “patch”, we apologize for the unclear text. We have added one sentence to clarify this: “*we subdivide the raw satellite image scenes into 336 by 336-pixel images (hereafter “patches”) as the input images for the model.*” (see page 4, lines 151-152).

References:

Brandt, M., Tucker, C.J., Kariryaa, A., Rasmussen, K., Abel, C., Small, J.L., Chave, J., Rasmussen, L.V., Hiernaux, P., Diouf, A.A., Kergoat, L., Mertz, O., Igel, C., Gieseke, F., Schöning, J., Li, S., Melocik, K.A., Meyer, J., Sinno, S.S., Romero, E., Glennie, E., Montagu, A., Dendoncker, M., & Fensholt, R. (2020). An unexpectedly large count of trees in the West African Sahara and Sahel. *Nature*, 1-5.

- p.5, l.173: "Fig. 1 shows the variation in the ensemble model arising from splitting the training dataset" – no it doesn't, there's virtually no differences visible between the three (random) splits shown in that figure.

Response: Agreed. We have deleted this sentence.

- p.5, l.177: "summarized" → "averaged"

Response: We have replaced “summarized” with “averaged” (see page 5, line 215).

- p.7, Fig.3: word has to be taken for granted that these red circles really are on wildlife individuals. This is very hard to see. Perhaps you can add (in other colours) false positives and false negatives? I find it hard to believe that the model made so little mistakes (cf. high precision and recall), given the minuscule individual size and background patterns that could be mistaken for animals at that scale.

Also, these red circles are generally visible, but in the bottom right figures they vanish when simulated with red-green-blindness vision. Maybe choose a more contrastive colour?

Response: The purpose of Fig. 3 is to show examples of the detected wildebeest (i.e., predicted result) across different landscapes with variation in wildebeest spatial clustering patterns using our deep learning pipeline, not to demonstrate the accuracy of our approach. The performance of our method was evaluated based on each wildebeest as demonstrated in Fig. 4, where the true positives, false positives and false negatives are all presented. Indeed, it is extremely challenging to visually and vividly depict the large-scale output of our study in the main text using figures given the minuscule individual size and complex background patterns. That is why we have also provided supplementary Figures 3-8 and animated images (a PowerPoint) along with the submitted manuscript, where readers can easily zoom in and out over large-scale, fine-resolution figures, and visually examine our results. Nevertheless, we agree that it is very hard to see the individual wildebeest in the zoomed-in squares in Fig. 3. Therefore, we have modified it and presented annotated and unannotated images side by side. We have enhanced the clarity of the animals and red circles in these figures. Please refer to the updated Fig. 6 in the revised manuscript.

- p.8, l.247ff: "Although the datasets were for different years" – I am wondering what this statement would tell to the reader. Detecting half a million individuals is impressive, but as you state this is a cumulative and thus highly inflated number. Looking at Table 1, the variance in detection count is enormous. Sure, much of this fluctuation can be attributed to different months of acquisition and climatic and other effects having occurred across the different years. It would have been great to see results across all years in imagery taken at the same time of the year each time.

Response: This comment is very important. We agree that the way we presented the numbers can be misleading. For clarification, we have revised this paragraph to “*The method resulted in a sum count of 480,362 (ranging between 470,121 and 490,603) individual wildebeest (F1-score: 84.75±0.18%) across the whole dataset (Table 1). See Fig. 7 for the location and coverage of the imagery of each year and Table 1 for the number of animals detected in each year.*” and moved it to the end of the last paragraph (see page 6, lines 259-263), and also removed the row of “sum count” in Table 1.

Regarding the suggestion of “It would have been great to see results across all years in imagery taken at the same time of the year each time”, as we can only use archived data for historical analysis, it is not easy to acquire satellite images taken at the same time for each year due to the limitation of data availability. Nevertheless, we are planning satellite image acquisitions for wildebeest census and it could be possible in the future to study the changes of migration patterns in different years and the relation to climate and habitat changes.

- p.8, l.256ff: follow-up: this is interesting, but one crucial aspect is missing: the discussion. Do these spatiotemporally characterised wildlife densities correspond to independent observations made about migration dynamics in the area?

Response: The most recently published survey data from the study area were collected by rear seat observers in piloted aircraft between 1977 and 2010 (Bhola, Nina, et al. 2012). Because the published data are summarized over decades and there is no temporal overlap with our satellite images, we cannot directly compare counts between the two. However, this long-term dataset provides a robust snapshot of historic trends in wildebeest distributions, and the spatial distribution of relative densities is comparable in areas of geographic overlap between the two datasets (see fig. 2b, g in Bhola et al. 2012).

Reference:

Bhola, N., Ogotu, J. O., Said, M. Y., Piepho, H. P., & Oloff, H. (2012). The distribution of large herbivore hotspots in relation to environmental and anthropogenic correlates in the Mara region of Kenya. *Journal of Animal Ecology*, 81(6), 1268-1287.

- p.11, Fig. 5: can you make these hotspot clusters semi-transparent? Why did you use a different background dataset than the images you already have? Also, I assume you used kernel density estimation to create these hotspots? You might want to at least put the term in the caption.

Response: Agreed. We have made these hotspot clusters semi-transparent as suggested. The reason we used a different background dataset is mainly for visualization purposes. Nevertheless, we agree that it is better to use the images we already have. Therefore, we have changed the background images. Please refer to the updated Fig. 8 in the revised manuscript (see page 29). In terms of the method used to create these hotspots (i.e., heatmaps), we used the Point Density tool instead of the Kernel Density tool in ArcGIS. The difference between the output of the Point Density and the Kernel Density is that in the point density, a neighborhood is specified that calculates the density of the population around each output cell. Kernel density spreads the known quantity of the population for each point out from the point location. We have indicated the method in the caption as suggested (see page 31, line 910).

- p.12, 1.323ff: "tremendous" spectral diversity is a bit of an overstatement, no? Also, this by itself is no warrant that generalisation to both new species and landscapes be feasible – if there's a domain shift it has to be accounted for; more variance in descriptors don't automatically make up for it. Furthermore, you explain the methods that you only keep the four "usual" bands (RGB-NIR), nullifying this advantage.

Response: Agreed. The use of “spectral diversity” is confusing here, and the word “tremendous” should have been avoided too. Regarding the generalization, we agree that for similar applications (such as wildebeest census at an even large scale in Serengeti-Mara), the current model could minimize the need to retrain the model with larger numbers of samples, while for new target species in new landscapes, the model will most probably need to be retrained with new samples. We have revised it to *“In addition to its size, the landscape diversity captured by this dataset will facilitate model transferability to applications in similar environmental contexts, such as future satellite-based wildebeest census surveys at the ecosystem scale. Although generalization of our model is inherently limited to wildebeest-like animals in open landscapes, the pipeline itself is generic and can be applied to other animal detection applications after retraining.”* (see page 7, lines 293-298).

- p.12, l.331f: here's the Achilles' heel of the proposed workflow: the method is (still) extremely expensive due to the requirement of high-resolution satellite imagery.

Response: We agree this method has historically been expensive for general users without funding or partnership with fine-resolution satellite imagery service companies. However, the era of the smallsat has disrupted the historically limited market for sub-meter satellite imagery, with at least 10 companies now offering sub-meter imaging capabilities from multiple constellations, at prices up to 90% less than traditional providers. In addition, there are increasing opportunities for researchers to access sub-meter imagery at low or no-cost through programs like NASA's Commercial Smallsat Data Acquisition Program, Planet's Nonprofit Program, or by partnering with any US Federal agency. Given the trajectory of these trends, we anticipate that satellite-based methods of counting large wildlife will become cost effective for researchers working in most open ecosystems within the next decade. We have added more explanation in the Discussion section: *"One limitation in satellite imaging wildlife currently is the cost of very-fine-resolution imagery. However, costs are falling as more companies are now offering sub-meter imaging capabilities from multiple constellations at lower prices. In addition, many satellite providers (e.g., Maxar and Planet) are providing more opportunities for researchers to access sub-meter imagery at low or no-cost."* (see page 8, lines 382-386).

- p.12, l.332ff: since you mention statistical methods on top of manual surveys (rightfully, as they are still the defacto standard for many conservation areas), it would have been interesting to include count estimations from them via spatially and temporally overlapping surveys. Do you have such data, at least for parts of the years/regions you included?

Response: We have attempted to link the satellite-based results with ground observations at the same time, such as GPS tracking data, balloon photographs or aerial surveys. However, due to the data availability limitation of historical satellite images as well as aerial/ground data, it is difficult to match the observations for further validation. Nevertheless, we are planning to conduct synchronized satellite image acquisition and aerial surveys in near future to match the observations.

- p.13, l.373: "this is an advancement" – again, playing devil's advocate I dare say this is a bit of an overstatement. Sure, object detectors struggle with such small animal sizes as mentioned, but the proposed method introduces other challenges (separation of two individuals close together isn't always guaranteed; number of clusters needs to be known a priori, etc.).

Response: We agree that object detectors could struggle with such small animal sizes. This is exactly the case for the wildebeest-sized objects with only a few pixels (< 9 pixels) covered in area. That was why we have decided to perform pixel-wise annotation and prediction, followed by individual disentangling by clustering. Nevertheless, we have conducted an experiment using a deep learning-based object detection algorithm (YOLOv4) on this study, and trained the model with the same dataset (same annotated points). The method resulted in an overall F1-score of 72.13% (with a precision of 76.95%, recall of 67.88%). Our method generates a better performance than the object detection algorithm even with varying image resolution, which clearly represents an advancement over prior work. Nevertheless, the research of small, weak

object detection is still far from complete, but given the breakthroughs over the past several years, we are optimistic about future developments.

We also agree that the separation of two individuals close together is not always guaranteed. The overall accuracy (F1-score) was 84.75%, not perfect yet. With respect to the number of clusters needing to be known *a priori*, we would like to clarify this. This is not really the number of clusters, but the size of our target animals presented in the satellite image. The wildebeest on these fine-resolution satellite images (38-50 cm) on average can be roughly represented by a 9-pixel segment, which can be inferred naturally based on *a priori* knowledge. If the target animal is much larger such as elephants, the standard object detector will work perfectly well since the features are captured more clearly on such spatial resolution.

- p.13, 1.378ff: given that it is already non-trivial to disentangle some species in very high-resolution UAV imagery, expecting this to be possible at a fine grain in satellite data is not realistic.

Response: Many thanks for this comment. We agree that the spatial resolution of satellite imagery is not sufficient to distinguish animal species. We also observed that general experience using aerial imaging in Mara shows that we would need to get down to a ground-sampling-distance of about 8 cm to be able to distinguish zebra from wildebeest. therefore, we have revised this sentence to: “*finer-resolution imagery (for example, <10 cm) will be required to discriminate these species.*” (see page 8, line 376).

- p.14, 1.428: does this mean that each animal was annotated with a 3x3 pixel square? I don't see the point in that (no pun intended).

Response: This segment size is designed to detect wildebeest-sized animals in the 38-50 cm resolution imagery. During annotation, each animal was labelled with a point at the centroid. Then the points were processed to pixel segments to be prepared for U-Net model training as it requires binary image masks: each segment is a 3 by 3 pixel square that is sufficient to cover the entire wildebeest in the vast majority of cases and pixels in this square are assigned a value of 1, whilst the remaining background pixels are assigned a value of 0.

- p.14, 1.429ff: doing this for inter-observer reliability is great for sure. I would be interested in both qualitative and quantitative measures on how high the agreement was: how big is the percentage of the intersection of the annotations the four experts agreed on, in comparison to all annotations made? How hard was it to spot the animals (the description of animal appearances are all good, but having experts' opinions on the feasibility of annotation would be quite helpful here, also given the subjectively high difficulty of the task shown in Extended Data Figure 2).

Response: Thank you very much for this insightful comment. The annotation of satellite images was carried out by four experts in two stages, with temporally different imagery being used independently for each annotator to identify and locate wildebeest animals through cross-referencing. Specifically, in the first stage, individual wildebeest in the satellite imagery were labelled manually by four annotators independently and separately supported by reference images. In the second stage, the initial annotations by the four annotators were combined

together into the final annotations via majority voting. For each labelled wildebeest in the first stage, it will be kept in the final annotations if three or more annotators agreed on this label. For these wildebeest that have been spotted by two annotators only, the four annotators will re-examine each case together based on the reference images and only add them to the final annotations if all four annotators agree on this. By performing such a two-stage process, the subjectivity was minimized and the level of human error in the final annotations was reduced. These labelled wildebeest are, therefore, used as ground reference for model training and testing with more confidence.

The level of agreement across the four annotators on each satellite image is listed in the Supplementary Table 1, where the respective amounts (and ratios) of wildebeest labels in the final annotations agreed by four annotators, three annotators (majority) and agreed after re-examinations are shown. Our overall percentage of agreement reached up to 96.1% (fully agreed independently), 3.5% (majority agreed independently), 0.4% (agreed jointly by re-examination).

Satellite Image	Wildebeest labels agreed by four annotators	Wildebeest labels agreed by only three annotators	Wildebeest labels agreed after re-examination
2009	20770 (94.1%)	1142 (5.2%)	153 (0.7%)
2010	8012 (98.0%)	159 (1.9%)	8 (0.1%)
2013	10284 (96.5%)	351 (3.3%)	24 (0.2%)
2015	621 (97.3%)	17 (2.7%)	0 (0.0%)
2018	10775 (95.4%)	414 (3.7%)	109 (0.9%)
2020	12419 (98.1%)	236 (1.9%)	6 (0.0%)
Overall	62881 (96.1%)	2319 (3.5%)	300 (0.4%)

- p.14, 1.439: was there a reason for this particular patch size?

Response: Many thanks for this question. Our patch size (336x336 pixels) was set by building a grid system, with each grid cell (image patch) of 150 m to 170 m depending on the spatial resolution of the satellite imagery (see page 9, lines 438-439). The patch size was designed to cover heterogeneous landscape characteristics and sufficient spatial context. Besides, we parameterized to this particular patch size based on the convolution layers of the U-Net model as well as computational power and capacity.

- p.14, 1.444ff: how much in percentage was the test dataset, then? This sounds like it is less than 0.15%, even though you later state (1.467) that it is actually 1.7%, again confirming a high spatial bias. Also, "stratified random sampling method" implies this is not the test, but the validation set – please specify.

Response: The testing set is approximately 1.7% of the whole dataset in terms of the area covered, and the number of animals in the testing set is about 2.4% of all the animals in the whole dataset (11,594 animals in total).

We disagree with the reviewer that this sample size confirms “a high spatial bias”. First, the volume and the animal sample size of test data included is arguably above average compared

with other existing similar studies (e.g. Brandt et al.: 100 sample plots in test set; Gonçalves et al.: 10,766 training points and 1,168 animals in test set). More importantly, even a small percentage of the population does not imply high spatial bias. Spatial bias will arise if preferential sampling has been used either knowingly or unknowingly. In this case, we aim to evaluate the performance of the model using a subset of the data, and the feature space of this subset (the test set) in general should be rich and representative of the whole image set, which is the reason we adopted stratified random sampling to make sure that the test set can well represent the whole dataset with great variation in terms of landscape and animal density, thereby largely reducing spatial bias in the test set (see page 10, lines 448-459). As a counter-example (which could be spatially biased), if we randomly took half of the whole image dataset as the test set, but this half happened to cover no wildebeest (it is obvious from Figure 7 and 8 that most of the area covers very small number of wildebeest), the accuracy would only reflect how good the model is to avoid false positives, but show nothing about the ability to accurately detect animals.

References:

Brandt, M., Tucker, C.J., Kariryaa, A., Rasmussen, K., Abel, C., Small, J.L., Chave, J., Rasmussen, L.V., Hiernaux, P., Diouf, A.A., Kergoat, L., Mertz, O., Igel, C., Gieseke, F., Schöning, J., Li, S., Melocik, K.A., Meyer, J., Sinno, S.S., Romero, E., Glennie, E., Montagu, A., Dendoncker, M., & Fensholt, R. (2020). An unexpectedly large count of trees in the West African Sahara and Sahel. *Nature*, 1-5.

Gonçalves, B. C., Spitzbart, B., & Lynch, H. J. (2020). SealNet: A fully-automated pack-ice seal detection pipeline for sub-meter satellite imagery. *Remote Sensing of Environment*, 239, 111617.

- p.15, l.489: maybe write "low-resolution but high-dimensional" to explain this rationale to non-expert readers.

Response: Agreed. Thank you. We have revised the text (see page 11, line 491).

- p.15, l.495ff: this is not clear enough: did you use a soft max activation and cross-entropy loss for classification (which is the norm) or did you apply a sigmoid activation and use binary cross-entropy losses instead?

Response: Thanks for this question. We used a sigmoid activation function (see page 11, line 498) and Tversky loss function (see page 11, lines 512).

- p.15, l.509: would it be possible to add an equation of the Tversky loss? That also makes it easier to understand the alpha and beta parameters.

Response: Yes. Thanks for your suggestion. We have added the equation in the supplementary information (see Supplementary Equation (1)).

- p.15f, l.517f: so the 2009 dataset serves as validation here? Why this particular year? Your entire train/val/test split setup seems increasingly confusing, to be honest.

Response: We apologize that we did not make it clear enough in the manuscript.

We would like to clarify that the training set of the year 2009 dataset is used for sensitivity analysis. The main purpose for this example (sensitivity analysis) is to explore how different settings of α and β can influence the performances of the individual U-Net models and the overall detection performance. We have rephrased it in the manuscript: “*We used the dataset of 2009 in a sensitivity analysis to evaluate how different settings of β influence the model performance*” (see page 11, lines 522-523).

There are two major reasons for using the 2009 dataset for sensitivity analysis. First, the 2009 dataset is the earliest satellite image we obtained for this study and the proposed ensemble learning model was initially designed based on this 2009 dataset. Second, the 2009 dataset captures the largest number of wildebeest with different spatial clustering patterns and covers various types of landscapes that are highly heterogeneous, which provides a better evaluation about how the proposed model performs in different conditions.

For the entire train/val/test split used in our main experiment (with the details reported in Table 2), we constructed the training and test sets from satellite images acquired over six years (2009, 2010, 2013, 2015, 2018, and 2020). The training set is composed of 1097 images with 53906 wildebeest captured. The test set is composed of 2700 images with 11594 wildebeest captured (see page 10, lines 443-446).

For model training, to enhance the diversity between the 10 U-net models, the training set is further split into ten folds. For each U-Net model, nine of the ten folds are selected to form the subset for model parameter training and the remaining one fold is used as the validation set for qualifying the performance during the training process. Thus, each U-net is trained with a different subset of training images (see page 11, lines 502-506).

- p.16, l.534f: how did you do this normalisation? Soft max? Sigmoid? Something else?

Response: We normalized the predictions to 0 to 1 using the standard min-max approach before averaging the results of all the 10 models. We have revised the sentence as: “*The probability map of each base model was first rescaled into the range of 0 to 1 (if the maximum value is greater than 0.05) and then averaged to obtain the final probability map as the output of the ensemble model*” (see page 12, line 539-541).

- p.16, l.537: it seems strange that the threshold chosen is precisely 0.5. In such an imbalanced setting I would have expected it to be different. Did you tune it on a validation set?

Response: The setting of the threshold (0.5 in this study) is a purely intuitive choice. Indeed, without any prior knowledge, the most commonly used threshold to binarize an arbitrary image with the pixel value range of [0, 1] is 0.5.

In this revision, we have performed a sensitivity analysis to study the influence of the value of this threshold on the wildebeest detection results based on the dataset of the year 2009. This sensitivity analysis is presented in Supplementary Fig. 10. It is demonstrated by this figure that

by setting the threshold value equal or very close to 0.5, one can maximize the F1 score of the detection result.

- p.18, 1.632ff: is there a reason why you did not publish your code on a dedicated site for open source projects like GitHub?

Response: We have published the code on GitHub (<https://github.com/zijing-w/Wildebeest-UNet>) (see page 13, line 634).

- Extended Data Fig. 3: this is a more interesting figure than Figure 2 in the main text, for reasons explained above (see minor comment): it also shows false positives and negatives. The difficulty of the scenes (esp. bottom row) is rather low, though.

Response: Thank you for this comment. We have moved this figure to the main article. Regarding the difficulty of the scenes, the intention to choose these three examples is to show the performance on different types of landscape, including heterogeneous and homogeneous environments. Nevertheless, we have updated this figure with new examples with more complex landscapes (see Figure 4).

- Extended Data Figure 6: this colour scale is... useless, really; there's no way one can identify performance differences this way.

Response: We have changed this figure to a table (see Supplementary Table 5).

- Extended Data Fig. 7: to be frank, tuning an evaluation criterion for maximum performance does not sound legit. It would have been better to motivate this choice by prior assumptions, e.g. on the expected average length of individuals, rather than to beautify model results by choosing the radius yielding highest performance readings.

Response: Agreed. The searching radius should be set based on the size of one pixel (see page 12, lines 546-548). We have removed the sensitivity analysis and the figure.

Reviewer #2 (Remarks to the Author):

This is an important paper in the growing field of counting large animals from space with satellite remote sensing. I encourage its publication.

The U-Net-based ensemble learning model approach outlined here represents (to my knowledge) an important next step in the remote sensing of animal abundance. It will likely prove useful in tracking the migrations of other large ungulates (pronghorn, caribou, bison, saiga, Tibetan gazelle, etc.). Maintaining these increasingly threatened migrations around the world is critical for continuing the ecological functioning of some of the last wild places on the planet. Getting reliable estimates of the abundance of keystone species on the move from space is a critical step for developing a more predictive ecology that will improve conservation and land management. Furthermore, it will do so in a manner that should be cheaper and less disruptive to the animals counted than current measures.

I found the paper to be quite thorough in terms of its description of the entire pipeline from satellite observations through model training and testing, validation and performance to postprocessing and detection. The discussion of model transferability both temporally and spatially was welcome. This is a fairly large study in terms of space (2747 sq. km.), time (images from 6 years across a total time frame of 12 years), total animals counted (~480,362), and the number of space sensors used (three satellites operating at different ground spatial resolutions). The figures are quite helpful (in addition to those in the paper, I found Extended Data Figures 1 and 2 to be especially useful). Their methods appear to be novel, sound, and sufficiently detailed.

Response: We really appreciate the remarks. Regarding the figures, we have moved the Extended Data Figure 1 and 2 to the main article (see Fig. 2 and Fig. 3).

One issue: Unless I'm missing something, I believe all of the training, ground reference, and validation data come from the same satellite datasets used in the analyses themselves. Given the spatial scales covered, I understand this. Nevertheless, I suspect that the community of ecologists and field biologists who have spent decades counting large ungulates on the move from ground counts and aerial platforms would like to see subsets of ground and/or aerial imagery used in the validation of the wildebeest counts published in this paper. I don't believe this constitutes a serious flaw in the analyses presented but do think future applications might consider how to incorporate samples from more traditional approaches for intercomparison with the results of the U-Net-based model. I realize scaling challenges will make this difficult but believe the links could still be made through incorporation of subset counts from more and less dense aggregations of wildebeest using traditional methods.

Response: We agree that "ground-truthing" insights from these satellite methods against traditional population survey techniques would be valuable. In fact, our team attempted to conduct such a ground-truthing exercise with the aim to align satellite imaging with conventional airplane survey transects spatially and temporally. Unfortunately, logistical constraints have thus far prevented alignment of this co-located imaging (e.g. cloud cover, flight timing restrictions). However, these types of validation/ground-truthing exercises would be an important future application using these models. Increased satellite constellation density and reduced cost of

imaging will certainly even further lower the bar for these kinds of follow-on studies in the near future. To note the value of this type of future research, we now include a new short paragraph highlighting this suggestion: “*A next valuable step in the science of enumerating large mammal populations using the proposed novel satellite-based methodology will be ground-truthing the predictions against both historical and contemporary estimates of population size derived using traditional methods (e.g., ground-based or aerial counts). For the present case of the wildebeest population, satellite-derived counts should be compared against the data collected every 2-3 years using aircraft surveys in the Serengeti National Park* ^{7,62}. Comparisons can be conducted both at the transect level (with satellite image acquisition synced to the timing of aircraft transects – although noting that temporal alignment of surveys with suitable conditions for both survey types can be challenging) and at the whole population level via data extrapolation.” (see page 7, lines 317-326).

A minor point: at a couple of places in the article wildebeest are referred to as “small-bodied” animals. I think animals of 1.5m to 2.5m don’t really qualify as small bodied—even among mammals.

Response: Agreed. Thanks for pointing it out. Wildebeest belong to large mammals, but the intention to use “small” in the paper is actually a relative comparison with other much larger mammals including elephants and whales mentioned in the previous literature. We have revised this description throughout the paper and changed “small-bodied”/ “small size” to “animals with a body length of 1.5-2.5 m” or “smaller” (see page 3, line 106; page 3, line 112).

There appears to be a minor typo in line 72 at the top of page 10 (in the caption for Figure 4) which refers to wildebeest detected in August 2019. I believe this should be August 2018.

Response: Thanks for pointing this out. We have revised the text accordingly (see page 29, line 888).

I like this work and wish NASA had funded it.

Response: Many thanks for supporting our work. Much appreciated.

Reviewer #3 (Remarks to the Author):

Review for: “Satellite-based monitoring of the world’s largest terrestrial mammal migration using deep learning”

This manuscript presents a machine learning approach to count individual ungulates in commercial satellite imagery. The approach seems to work well across imagery from different satellites, with slightly different spatial resolutions, across distinctive habitats – which is impressive. Overall, this work presents an important first step in developing approaches to leverage commercial satellite imagery for monitoring wildlife populations. I have expertise in animal movement and migration ecology, but I am not an expert in machine learning. Therefore, I have prepared my review as a potential user of this approach and cannot speak to the technical soundness of the approach presented in the manuscript. I was enthusiastic to read this manuscript, because this technique would be useful and relevant in my research. However, I do have some concerns about the manuscript and the utility of this approach for monitoring wildlife and understanding animal behavior.

Main concerns/points to consider:

Essentially, this is a proof of concept that commercial satellite imagery can be used to count herds of ungulates. The main outcomes of this approach are 1) counts derived from imagery on 6 days during different parts of the migration cycle, 2) heatmaps of ungulate density and 3) the analytical pipeline. No further analyses are done and no further ecological insights are drawn from this data. So, as I see it, the key contribution of this manuscript is the development of a new method to count wildlife using submeter-resolution satellite imagery. Yet, although the main contribution is methodological, the format of Nature Communications presents all methodology details at the very end of the manuscript and within the supplements. This misalignment of the core contribution of the paper and the formatting requirements of Nature Communication seems odd and potentially problematic. For this reason, Nature Communications might not be an ideal fit for this manuscript. However, I will leave this up to the editor to determine. I think there is wide enough interest in this topic, that it could warrant publication in a high-impact journal. I am just not sure that the methods last format of Nature Communications is logical for this contribution.

Response: Many thanks for this extremely helpful summary and comments. We have followed the suggestion of the editor and added some of the key methodological explanations of the study to the start of the Results section (see page 4, lines 147-169).

Next, how were the dates of the imagery chosen? The selection of August 2009, September 2010, August 2013, July 2015, August 2018, and October 2020 seems somewhat random. Is this because those were the dates where images were readily available within your area of interest or was there some other reason for picking these dates? More information on temporal resolution and availability of data is needed to assess the utility of this method for broad application in ecology and wildlife monitoring. It would be useful to know if the temporal resolution of image availability is a limiting factor, because this would affect the types of questions researchers can address and how widespread and useful this new method would be.

Response: The selection of the images used in our study was not completely random. It depends partly on the availability of the archived images and partly on the research questions we are asking. Our study aims to investigate whether the state-of-the-art deep learning algorithm can be used to accurately and automatically locate and count large herds of migratory ungulates across a large and heterogeneous area in the Serengeti-Mara ecosystem using fine-resolution satellite imagery, and consequently quantify the variation in the wildebeest aggregation patterns across space and time. Additionally, we would like to examine if our approach works across different satellite sensors with various spatial resolutions. For this purpose, we decided to focus on the dry-season range (from mid-July to mid-October) of the migratory wildebeest in the Masai Mara National Reserve and the northern part of the Serengeti National Park, where migratory animals are expected to be highly aggregated and more cloud-free satellite images are likely to be available. Based on these requirements, we firstly preview the fine-resolution satellite imagery available on Google Earth, where we can visually check if the large herds of migratory ungulates are present. Then we further check the spatial resolution of these images at <https://discover.maxar.com/> (a website for searching and discovering the archived high-resolution satellite images collected by Maxar satellite constellation). Finally, we select and order these images for our study.

Regarding the question about the temporal resolution and availability of data, unlike the free and coarse resolution Earth Observation satellites (e.g., Landsat and Sentinel-2), which regularly collect images with a temporal resolution of 16 days (Landsat) or 5 days (Sentinel-2), most commercial operators e.g., Maxar, collect imagery over a specific area at a specific date on request or opportunistically (e.g., when a civil war erupts they start to collect imagery as there is likelihood of sale) so there is not regular coverage of an area at fixed intervals. In regard to the temporal resolution, for example, the Maxar constellation (specifically GeoEye-1, WorldView 2 and 3) can collect more than 3.8 million square kilometers per day and has a revisit rate of up to 1-2 times per day. Thus, the collection of high temporal resolution commercial satellite images is not an issue if the weather conditions (i.e., cloud cover) allow. We have added one additional sentence in the corresponding paragraph: “*We selected these images from the archived very-fine-resolution satellite images acquired by the Maxar Worldview constellation, which can cover more than 3.8 million square kilometers per day and has a revisit rate of 1-2 times per day.*” (see page 9, lines 406-409).

In our study, we used the archived very-fine-resolution satellite images to demonstrate method development and large-scale applications. However, the selection of the images is somehow limited by the data availability. We have checked the achieved Maxar satellite images at <https://discover.maxar.com/>, and there are more than 150 images available for Masai Mara National Reserve and Serengeti National Park with various coverages (from tens of square kilometers to thousands of square kilometers) between 1999 and 2022.

How ready to use is the software/code in Dryad? A main contribution of the work is the analytical pipeline, so having usable code would be a huge contribution. I couldn't tell, because the link to the dryad repository isn't active/accessible yet.

Response: We realized the link to the code in Dryad is not accessible because the pdf reader does not recognize a link that spans over two lines. We have uploaded the code to GitHub and added the new link (<https://github.com/zijing-w/Wildebeest-UNet>) (see page 13, line 634). The code can be directly implemented using Google Colaboratory (<https://colab.research.google.com/>) without complicated installation or configuration. However, the complete satellite imagery dataset cannot be provided due to the restrictions of data sharing policy. The user can use a relatively small sample dataset to train and test the model.

Transferability How transferable is this pipeline to other datasets? In lines 323-325 of the discussion, it is mentioned that the scope and size of this project makes it so that this model could be easily generalized to new systems and species. I think anyone interested in generalizing this model to their system would be curious to know how involved this process would be. For example, to use this approach on caribou in the artic, would you need to annotate new training data? How much? How would you recommend iterating and tweaking the model? This type of information would be useful to potential end-users looking to adopt your approach.

Response: We realized the original texts can be misleading and thus have updated these lines to: *“In addition to its size, the landscape diversity captured by this dataset will facilitate model transferability to applications in similar environmental contexts, such as future satellite-based wildebeest census surveys at the ecosystem scale. Although generalization of our model is inherently limited to wildebeest-like animals in open landscapes, the pipeline itself is generic and can be applied to other animal detection applications after retraining.”* (see page 7, lines 293-298).

More specifically, for applications that share similar environmental contexts, such as wildebeest census in other parts of Serengeti-Mara or other species in the same ecosystem, the trained model can provide a good start for new tasks with a relatively small number of new samples. For a completely different application such as a new species in a new system (like caribou in the arctic), a new model will need to be initialized and trained with new data, which means we need to annotate new samples of the animal. First, depending on the size of the animal on the imagery, the size of one animal segment needs to be updated. For example, if the animal covers approximately 9 pixels on the images, we can continue with the default settings. If the animal is smaller, e.g., 4-pixels large, we need to update this number in the code. Second, the size of the samples will depend on how large and how heterogeneous the landscape is. We recommend to start with a small training and validation dataset (e.g., 100 image patches) and then add more training samples according to the performance on the validation set.

Validation/labeling data – how consistent were the 4 annotators that labeled the training dataset? I am curious to know if the level of human error is similar to that of the deep learning algorithm? Since there is no independent way to validate the counts (besides using human-annotated data from the same commercial images), I think more exploration could be warranted.

Response: Thank you for this constructive comment on validation/labelling data. Our labelling procedure was structured into two stages, with temporally different imagery being used independently for each annotator to identify and locate wildebeest through cross-referencing.

The first stage involves labelling of individual wildebeest on the satellite imagery by four annotators independently and separately supported by reference images. The second stage combined the initial annotations by the four annotators into final annotations via majority voting (>3 annotators agreed consistently). For these wildebeest that have been spotted by two annotators only, the four annotators re-examined each case together based on the reference images to agree on the final annotations jointly. In this way, the level of human error is significantly reduced.

The level of consistency/agreement across the four annotators on each satellite image is listed in Supplementary Table 1, where the respective amounts (and ratios) of wildebeest labels in the final annotations agreed by four annotators, three annotators (majority) and agreed after re-examinations are shown. Our overall percentage of agreement reached up to 96.1% (fully agreed independently), 3.5% (majority agreed independently), 0.4% (agreed jointly by re-examination).

Satellite Image	Wildebeest labels agreed by four annotators	Wildebeest labels agreed by only three annotators	Wildebeest labels agreed after re-examination
2009	20770 (94.1%)	1142 (5.2%)	153 (0.7%)
2010	8012 (98.0%)	159 (1.9%)	8 (0.1%)
2013	10284 (96.5%)	351 (3.3%)	24 (0.2%)
2015	621 (97.3%)	17 (2.7%)	0 (0.0%)
2018	10775 (95.4%)	414 (3.7%)	109 (0.9%)
2020	12419 (98.1%)	236 (1.9%)	6 (0.0%)
Overall	62881 (96.1%)	2319 (3.5%)	300 (0.4%)

How useful would this approach be for monitoring a migratory population if the migration pattern is unknown? In many places, where animals move is not well documented. And as we place more tracking devices on animals, scientists are frequently discovering more surprising new migration routes. How do you see this approach fitting in more broadly with the use of other tracking technologies?

Response: We have added one paragraph discussing the potential application of this approach together with GPS tracking technologies on studying unknown migration patterns: *“Another potentially promising application of the proposed method would be the detection of large mammal migrations that have not previously been documented. Despite the charisma of such fauna, the migrations can go uncharacterized and are infrequently discovered or rediscovered (e.g., the Burchell’s zebra migration in Namibia/Botswana⁶⁵; white-eared kob in South Sudan⁶⁶). Given the advantages of surveying at large scales, satellite imaging techniques, coupled with GPS tracking of individual animals, could provide a powerful methodological combination for detecting or confirming such migrations. GPS tracking data could benefit the survey by giving prior information about the potential range, while regularly acquired satellite imagery can be used to identify the migration routes of large animal groups over time, as satellite imaging at high time frequency becomes possible. Such methods are also especially useful for detecting and studying migrations in remote or insecure regions⁶⁶.”* (see page 8, lines 344-355).

Minor points:

I found it slightly confusing that in the abstract you say that the method counts large herds of migratory ungulates (wildebeest and zebras). Then, all of the figures and results are labeled specifically as wildebeest detections. Finally, when we move back into the discussion section, there is an entire paragraph that acknowledges that it is not possible to distinguish between wildebeest and other similarly sized ungulates. So I wonder, if the way the results are labeled and presented should be modified to reflect this uncertainty.

Response: We appreciate this observation and indeed, the method detects similar-sized wildebeest and zebras together. But since wildebeest is the dominant species, for simplicity reasons we refer to the detected animals as wildebeest in the Results and Methods sections. We have added a sentence to clarify this: *“Note that as wildebeest is the dominant ungulate species in the system and most animals we located and counted were wildebeest, we refer hereafter to the migratory ungulates detected by our model as wildebeest for the purpose of simplicity.”* (see page 4, lines 169-172).

Lines 109-113 of the introduction – What exactly do you mean by “this technology”. Is it the automated approach to counting wildlife using commercial satellite images? Or the ability to frequently and accurately assess the status of migratory ungulate populations? I think there are still too many dots to connect between an automated approach to counting animals in satellite imagery and understanding emergent properties of animal groups. For example, I think you would need more frequent data collection, and the ability to differentiate between individuals across time periods to truly get at any of these emergent phenomena. I suggest either deleting this sentence or pairing it back to be more relevant to the ecological questions your analytical pipeline can help to tackle now.

Response: We appreciate this thoughtful feedback and have updated the main text (see page 3, lines 133-134) to clarify that we are referring to the automated approach of mapping large groups of animals across the landscape via satellite imagery (i.e., the new methodology, not technology). While we fully agree that more frequent imagery and the ability to track individuals over time would rapidly advance understanding of the emergent properties of animal groups, we also advocate that there is a long history of describing the emergent properties of ecological patterns from discrete, point-based datasets in ecology. For example, we believe there is much to be learned from the fields of entomology and plant science, which have extensive bodies of literature regarding the use of spatial point process theory to describe emergent properties of spatial patterns observed in plant and insect communities (e.g., Law et al. 2009; Vinatier et al. 2011; Pringle and Tarnita 2017). Similarly, the locations of hundreds of thousands of wildebeest can readily be analyzed as a spatial point process that can then be related to both social (relative locations of other animals) and environmental (landscape features) drivers of occurrence at multiple scales (individual to group).

Moreover, preliminary research from co-authors Hughey and McCauley has found predictive links between the spatial structure of wildebeest groups (i.e., relative orientation, spacing, and alignment of individuals) and the behavior of animals on the ground. The findings of this

forthcoming work imply our assertions about the potential for identifying mechanisms of behavioral responses from satellite-derived animal detections.

To clarify this in the main text, we have added two references below as an example of the type of analysis that could be done to assess the emergent properties of animal aggregations from satellite images. However, we have chosen to leave the reference to understanding mechanisms to protect the novelty of work in progress.

References:

Law, Richard, Janine Illian, David F. R. P. Burslem, Georg Gratzer, C. V. S. Gunatilleke, and I. A. U. N. Gunatilleke. 2009. "Ecological Information from Spatial Patterns of Plants: Insights from Point Process Theory." *The Journal of Ecology* 97 (4): 616–28.

Pringle, Robert M., and Corina E. Tarnita. 2017. "Spatial Self-Organization of Ecosystems: Integrating Multiple Mechanisms of Regular-Pattern Formation." *Annual Review of Entomology* 62 (January): 359–77.

Vinatier, Fabrice, Philippe Tixier, Pierre-François Duyck, and Françoise Lescouret. 2011. "Factors and Mechanisms Explaining Spatial Heterogeneity: A Review of Methods for Insect Populations." *Methods in Ecology and Evolution*. <https://doi.org/10.1111/j.2041-210x.2010.00059.x>.

Figure 3 – is it possible to include the zoomed-in squares in Fig.3 A-E without the red circles? It is hard to see what the algorithm is picking up on with the image modified like this. I think having both the annotated and unannotated images side by side would be a helpful visualization.

Response: Yes, it is possible to include the zoomed-in squares in Fig.3 A-E without the red circles. We have modified it and presented annotated and unannotated images side by side as suggested. Please refer to the updated Fig. 6 in the revised manuscript.

REVIEWERS' COMMENTS

Reviewer #1 (Remarks to the Author):

I would like to thank the authors for the detailed and constructive response letter, additionally conducted experiments and clarifications. I particularly appreciate the responses of disagreement with my statements; they are well-motivated and a good scientific discussion (in particular about the spatial bias; upon re-reading the paper and comment I can follow the motivation). Upon reading through the response I also realised a stray minor comment from my side (with "TODO" in it) I wanted to revise upon complete review but overlooked, along with my own confusion about the loss function used. Apologies for that.

The manuscript itself now has a lot more ecological embedding, which is not only important for a journal like this, but also helps reach a broader target audience.

MAJOR COMMENTS:

- placement of method description (page 4, lines 145-172): this is a bit of a difficult one, given the situation. I personally believe this high-level description of the method should go straight into the "Methods" section at the end, not the "Results" section. Of course, from Reviewer #3's comment I understand why you decided to place it here and see your conclusion that the methodology is the key novelty. I personally disagree with this latter claim. Your methodology is new from an application point of view, not from a methodological one. This does not deprive it from value; on the contrary. For me, the biggest contribution in your work is the large-scale application, study of effects and (now great) comparisons, and implications and embedding in the greater context. Hence, putting a description of how a U-net works into the spotlight deflects attention from the actual value of your study.

I will leave it up to you and the editor to decide. This is a well-written paragraph, it is informative and important to have. It's just its placement I am not sure about.

- about YOLOv4: I highly appreciate the comparison you made to it. I see that you have referenced it in the text, but only for a motivational comment about object sizes. You did not seem to have reported the results outside the response letter, not even in the supplementary information document. I would recommend including this, not only to justify your segmentation followed by clustering approach, but also to provide more information for those readers who know object detection models.

MINOR COMMENTS:

- p.3, l.94: I'd add the original reference to "Faster R-CNN" (if you want to keep this as an example at all). Same line: performance suffers from small object sizes, not the detectors

- p.3, l.98: maybe add a typical resolution for satellite imagery to put the relationship between animals and pixels into perspective

- p.3, l.101ff: redundancy regarding U-net

- p.3, l.113ff: I would very slightly reword this and put more emphasis on the postprocessing (this is to reflect the limitations about a vanilla U-net you mentioned in the paragraph before)

Reviewer #2 (Remarks to the Author):

The concerns expressed in my initial review have been addressed by the authors' revisions to the manuscript. I recommend publication.

Reviewer #3 (Remarks to the Author):

The authors did a nice job with the revisions and I have no further comments or concerns.

Reviewer #1 (Remarks to the Author):

I would like to thank the authors for the detailed and constructive response letter, additionally conducted experiments and clarifications. I particularly appreciate the responses of disagreement with my statements; they are well-motivated and a good scientific discussion (in particular about the spatial bias; upon re-reading the paper and comment I can follow the motivation).

Upon reading through the response I also realized a stray minor comment from my side (with "TODO" in it) I wanted to revise upon complete review but overlooked, along with my own confusion about the loss function used. Apologies for that.

The manuscript itself now has a lot more ecological embedding, which is not only important for a journal like this, but also helps reach a broader target audience.

Response: Thank you very much for reviewing, recognizing, and supporting our work. Your valuable, constructive comments have helped us to further improve the quality of this paper.

MAJOR COMMENTS:

- placement of method description (page 4, lines 145-172): this is a bit of a difficult one, given the situation. I personally believe this high-level description of the method should go straight into the "Methods" section at the end, not the "Results" section. Of course, from Reviewer #3's comment I understand why you decided to place it here and see your conclusion that the methodology is the key novelty. I personally disagree with this latter claim. Your methodology is new from an application point of view, not from a methodological one. This does not deprive it from value; on the contrary. For me, the biggest contribution in your work is the large-scale application, study of effects and (now great) comparisons, and implications and embedding in the greater context. Hence, putting a description of how a U-net works into the spotlight deflects attention from the actual value of your study.

I will leave it up to you and the editor to decide. This is a well-written paragraph, it is informative and important to have. It's just its placement I am not sure about.

Response: Thank you for this suggestion. We prefer to keep it as it stands and let the editor decide.

- about YOLOv4: I highly appreciate the comparison you made to it. I see that you have referenced it in the text, but only for a motivational comment about object sizes. You did not seem to have reported the results outside the response letter, not even in the supplementary information document. I would recommend including this, not only to justify your segmentation followed by clustering approach, but also to provide more information for those readers who know object detection models.

Response: Thank you for this suggestion. After careful thought and discussion with our co-authors, we decided not to include the YOLOv4 results in the manuscript due to consideration of the focus and logic of our paper. However, we propose a solution below

which we hope you find acceptable.

The main objective of the paper is to present a transferable deep learning pipeline for large-scale application, instead of comparing different methods. Our study accomplished that goal and demonstrated the effectiveness of the proposed approach.

Nevertheless, we present a table in this response letter to show a more detailed result derived from the YOLOv4 experiment (Table R1). The YOLOv4 code was obtained from https://github.com/WongKinYiu/PyTorch_YOLOv4. We trained the YOLOv4 model with the same dataset. **Note:** as part of the transparent peer review policy imposed by the *Nature* journal, we have agreed to the publication of the reviewer comments to the authors and the author rebuttal letter of revised versions as a supplementary “peer review file”. In this way, any potential readers who are interested in the performance of the object detection models (i.e., YOLOv4) can find it here. We hope that you are happy with this solution.

Table R1: The wildebeest detection accuracy of the YOLOv4 model for each of the six years and the whole dataset

Satellite	Acquisition date	Spatial resolution	Precision	Recall	F1-score
GE01	11/Aug/2009	43 cm	78.00%	75.68%	76.82%
GE01	24/Sep/2010	44 cm	81.41%	65.34%	72.50%
GE01	10/Aug/2013	45~50 cm	71.77%	58.82%	64.65%
WV03	17/Jul/2015	38 cm	82.20%	77.43%	79.74%
GE01	02/Aug/2018	42~48 cm	75.34%	62.93%	68.58%
WV02	08/Oct/2020	50 cm	76.29%	66.59%	71.11%
Overall			76.95%	67.88%	72.13%

MINOR COMMENTS:

- p.3, l.94: I'd add the original reference to "Faster R-CNN" (if you want to keep this as an example at all). Same line: performance suffers from small object sizes, not the detectors

Response: We have removed the example of Faster R-CNN. We agree with the correction you made to the other sentence and have revised it to “... *using object detection algorithms. However, the performance of current object detectors suffers from the small size of the objects in imagery*³⁹⁻⁴¹.” (see Lines 93-94).

- p.3, l.98: maybe add a typical resolution for satellite imagery to put the relationship between animals and pixels into perspective

Response: Agreed. A typical spatial resolution has been added here and now it reads as

follows: “... *both of which have more than eight pixels along the body length axis in submeter-resolution (e.g., 0.3-0.5 m) satellite imagery.*” (see Line 98).

- p.3, l.101ff: redundancy regarding U-net

Response: Indeed, you are right. It is redundant to use the words "like U-Net" here. We have removed it from our manuscript as you suggested (see Line 102).

- p.3, l.113ff: I would very slightly reword this and put more emphasis on the postprocessing (this is to reflect the limitations about a vanilla U-net you mentioned in the paragraph before)

Response: Agreed. We have revised it to “We do this by integrating a post-processing clustering module with a U-Net-based deep learning model, which uses high-precision pixel-based image segmentation to locate animals at the object level.” as you suggested (see Lines 114-115).

Reviewer #2 (Remarks to the Author):

The concerns expressed in my initial review have been addressed by the authors' revisions to the manuscript. I recommend publication.

Response: Thank you very much for your recognition and support of our work.

Reviewer #3 (Remarks to the Author):

The authors did a nice job with the revisions and I have no further comments or concerns.

Response: We appreciate your recognition and support of our work.